# Last Interglacial sea-level proxies in East Africa and the Western Indian Ocean

Patrick Boyden[1], Jennifer Weil-Accardo[2], Pierre Deschamps[2], Davide Oppo[3], and Alessio Rovere[1]

[1]MARUM - Center for Marine Environmental Sciences, University of Bremen, Germany
[2]Aix Marseille Université, CNRS, IRD, Collège de France, CEREGE, France
[3]Sedimentary Basins Research Group, School of Geosciences, University of Louisiana at Lafayette, USA

**Correspondence:** Patrick Boyden (pboyden@marum.de)

**Abstract.** In this paper, we describe a sea-level database compiled using published Last Interglacial, Marine Isotopic Stage 5 (MIS 5), geological sea-level proxies within Eastern Africa and the Western Indian Ocean (EAWIO). Encompassing vast tropical coastlines and coralline islands, this region has many occurrences of well preserved last interglacial stratigraphies. Most notably, islands almost entirely composed of Pleistocene reefs (such as Aldabra, the Seychelles) have provided reliable paleo relative sea-level indicators and well-preserved samples for U-Series chronology. Other sea-level proxies include uplifted marine terraces in the north of Somalia and Pleistocene aeolian deposits notched by MIS 5 sea level in Mozambique to tidal notches in luminescence limited aeolian deposits in Mozambique. Our database has been compiled using the World Atlas of Last Interglacial Shorelines (WALIS) interface and contains 58 sea-level indicators and 2 terrestrial limiting data points. The database is available open access at https://doi.org/10.5281/zenodo.4043366 (Version 1.03, Boyden et al., 2020).

## 1 Introduction

The main aim of this paper is to describe a standardized database of geological sea-level proxies, compiled using the tools available through the World Atlas of Last Interglacial Shorelines (WALIS) project (https://warmcoasts.eu/world-atlas.html). WALIS includes an interface that can be used to organize relative paleo sea-level data into a standardized data framework. Once saved in the interface, the data is stored in a MySQL database and can be exported as a multi-sheet spreadsheet that contains data and metadata on sea-level proxies. The exported spreadsheet for the Eastern Africa and Western Indian Ocean region (EAWIO) is the subject of this paper and is available open access here: https://doi.org/10.5281/zenodo.4043366 (Version 1.03, Boyden et al., 2020). In addition, we have included 21 U-Series ages compiled within WALIS by Chutcharavan and Dutton (2020).

Pleistocene sea-level records for the coast of Eastern Africa and the Western Indian Ocean (EAWIO) were first described in 1894 by British naturalist William Abbott. During an expedition onboard *H.M.S. Alert* in 1882, Abbott surveyed the Aldabra and Glorieuses islands groups, providing the first description of raised coral reefs in this area. It would not be until the 1920s and 1930s before the coastal geomorphology of EAWIO was revisited and new sites were added.

In memory of the extensive contributions made by René Battistini (1927-2017) throughout the EAWIO region, in this paper we will use his proposed *Aepyornien* (Malagasy Quaternary) nomenclature where appropriate (Battistini, 1984). According to

classic definitions, the Aepyornian is punctuated by three major marine transgressions: Tatsimian (MIS 7), Karimbolian (MIS 5e), and Flandrian (Holocene — originally from the Netherlands) (Figure 1). Guilcher (1954) was the first to describe emerged fossil reefs at 15 m above sea level, which he assumed to be Pliocene or Quaternary, on the northern tip of Madagascar, near the town of Antsiranana. Guilcher (1956) later described two separate emerged reefs at 4 m and 12 m above mean sea level (MSL) on the nearby Orangea peninsula.

During the 1960s and 1970s, the advent of U/Th dating using Alpha-Spectrometry spurred on a new wave of publications in the EAWIO. While these publications offer invaluable information regarding local morphology and stratigraphy, analytical limitations along with focuses deviating from sea-level research prevent many of these early studies from being included within WALIS. For example, Guilcher (1954) and Battistini (1965a) provided two of the first morphological and chronological descriptions of northeastern Madagascar, but it was not until Stephenson et al. (2019) that more accurate elevation and chronological data would be gathered.

In the mid-1950s and 1960s, the Geological Survey of Kenya undertook successive campaigns to map and describe coastal geology (Caswell and Baker, 1953; Thompson, 1956; Williams, 1962). Caswell and Baker (1953) in particular described two marine transgressions along the Kenyan coast, resulting in a succession of coral reef terraces. Battistini (1969) followed with a description of the two most recent marine transgressions in the vicinity of Mombasa and Malindi, with additional detailed stratigraphy of the reef terraces conducted by Crame (1980). Here, the MIS 5e shoreline is primarily dominated by erosional benches, exposing the back reef lagoonal sediments of the Tatsimian. The emphasis then shifted to the ecology of these paleo reef environments, and away from paleo-sea level reconstructions (Crame, 1986).

Similarly, in Tanzania, early attempts were made to describe the coastal geomorphology. Stockley (1928) was the first to describe emergent reefs as a dominant lithology of the Zanzibar archipelago. This was followed much later by Battistini (1966) who described several cropping out reefs along the central Tanzanian coast, which he attributed to the same transgressive sequences he observed on the northern coast of Madagascar (Tatsimian and Karimbolian). Northwards towards the border with Kenya, Alexander (1969) described a series of emergent, well-developed beach ridges forming three distinct groups, unfortunately, no concrete age was established. Cooke (1974) and Adey (1978) both provide additional stratigraphic descriptions of emergent reef facies on the Tanzanian and Zanzibarian coasts respectively. Again, however, chronological data is lacking and only stratigraphic constraints were provided.

After a long hiatus Abbott (1894), Stoddart (1967) began a series of expeditions to the outer, isolated islands of the Seychelles. These small islands like Assumption, Cosmoledo, and Astove, are primarily made of emergent reef deposits around 4-5 m above MSL (Bayne et al., 1970a, b; Korotky et al., 1992). Unfortunately, no chronological constraints were produced during these expeditions. In the Granitic Seychelles, Montaggioni and Hoang (1988) provided the first sea-level specific survey for emerged coralline outcrops adhered to the granitic basement rock with U-Series Alpha-Spectrometry ages ranging in elevation from 2 to 7 m above MSL.

Far to the south, Camoin et al. (1997); Montaggioni (1972, 1974, 1976, 1982) provide detailed descriptions of the Mascarene Archipelago, home to Mauritius and Reunion Islands. While mainly volcanic in origin, the Mascarene Archipelago has extensive modern fringing coral reefs and a few occurrences of paleo emerged reefs (Faure, 1977). Battistini et al. (1976) first

described emergent Pleistocene reef sections along the western coast of Mauritius, but for the most part, post-volcanic subsidence means that the majority of Pleistocene outcrops are either covered by more recent aeolian sands or Holocene modern coral accumulation (Camoin et al., 1997).

## 2 Methods

### 2.1 Types of Sea Level Indicators

Within the EAWIO region, we identified six main types of sea-level indicators: coral reef terraces, lagoonal deposits, marine terraces, shallow-water coral reef facies, tidal inlet facies, and tidal notches. As the region is situated within the tropics and sub-tropics, coralline related indicators are among the most studied as well as the best chronologically constrained. In Table 1, each indicator's indicative range (IR) and reference water level (RWL) are described. The IR is defined by the upper and lower limits of where the indicator forms in relation to a known datum (e.g. MSL), the RWL is the mid-point of the IR. These two

values define the indicative meaning for each sea-level proxy, which is used to define where the paleo sea level was located with respect to the measured position of the landform (Shennan, 1982; Van de Plassche, 2013; Shennan et al., 2015). Additionally, two dune deposits are included as terrestrial limiting points within WALIS. For these two points, it can be only determined that sea level was, at the time of their formation, below the measured elevation of the landform.

### 2.2 Surveying Techniques

Very few studies within the EAWIO have the express intent to establish detailed surveys of Last Interglacial (LIG) sea-level proxies. This is especially true with respect to elevation measurements. Most surveys conducted during the 20[th] century do not report a methodology used in measuring elevations. It is not until the advent of Global Navigation Satellite Systems (GNSS) and Total Stations that surveys on many of these remote shorelines could be accurately documented. The elevation measurement techniques used in the studies that we compiled in the database are shown in Table 2. When no accuracy was given for an

elevation measurement in the original study, the typical accuracy of the technique was used. Any elevation measurement must be related to a specific sea-level datum (Table 3). Unfortunately, in the literature we surveyed, it was often unclear how most datums were established (e.g. how the highest tide level was calculated at different sites). Instead authors will often state that the elevation is relative to mean sea-level or the level of highest seas. This uncertainty is exacerbated by the large variance in tides within the EAWIO, specifically in the immediate vicinity of the Mozambique Channel (Farrow and Brander, 1971;

Kench, 1998). In the database, we therefore try to reflect this uncertainty within the elevation measurement for each proxy.

### 2.3 Dating Techniques

Early observations of paleo-shorelines relied primarily on chronostratigraphic constraints to try and piece together a regional narrative. Two formations are primarily used in early studies: the Aldabra Limestone (Aldabra, Seychelles) and the Karimbolian Limestone (Antsiranana, Madagascar). The Aldabra Limestone is characterized by reef limestones with large corals in

growth position. Similarly, the Karimbolian Limestone, first described by Guilcher (1956), refers to massive reefs overlain by red aeolianites. Both of these formations have since been chronologically constrained using U-Series Alpha-spectrometry (Thomson and Walton (1972) and Battistini and Cremers (1972) respectively).

As with elevation measurement techniques, dating techniques within the EAWIO have advanced dramatically since the first chronologies became available in the early 1960s, thanks to U-Series ages from coral samples (Barnes et al., 1956; Thurber et al., 1965). In general terms, U-Series ages are derived by measuring the disequilibria between $^{238}$U, $^{234}$U, and $^{230}$Th radioisotopes (Edwards et al., 2003). The reliability of $^{230}$Th-ages relies on a closed-system behavior that can be compromised by post-depositional processes that lead a re-mobilization of uranium –or thorium- within the coral skeleton. Several mineralogical and isotopic screening criteria are generally applied to detect any opening of the $^{230}$Th-$^{234}$-$^{238}$U system. The coral samples should show no evidence of diagenetic alterations such as recrystallization or transformation of primary aragonite to secondary calcite. This is generally assessed by quantification of secondary calcite. In most recent studies, coral samples showing a calcite content of more than 1% are usually discarded. The uranium content of fossil corals should ordinarily be similar to modern ones (about 2.8 ppm). The back-calculated $[^{234}U/^{238}U]_0$ ratio that represents the $[^{234}U/^{238}U]$ ratio at the time of coral growth should reflect the $[^{234}U/^{238}U]$ of seawater. Due to the long oceanic residence time of uranium, this ratio is supposed to be similar to the modern seawater. For this reason, fossil corals showing a $[^{234}U/^{238}U]_0$ significantly different from modern seawater were generally discarded (Hamelin et al., 1991; Bard et al., 1991). This isotopic criterion is not always strictly applied since 1) there are many evidences that the $[^{234}U/^{238}U]$ ratio of seawater may have varied through time (see discussion in Chutcharavan et al., 2018) and 2) some models, assuming decay-dependent redistribution of $^{234}$Th and $^{230}$Th were developed to correct for the "open-system" behavior highlighted by anomalous $[^{234}U/^{238}U]_0$ ratios (Thompson et al., 2003; Villemant and Feuillet, 2003). Although these open-system ages are questionable, some of the ages reported here are calculated using such model (Thompson et al., 2003). Here, we state whether if ages were originally reported as closed- or open-systems. It is important to note that the application of the screening criteria presented here are quite recent and that most of the studies reported in this review were carried out before these criteria became common practice in the U-Th community.

Precision of U-Series ages relies upon the analytical method used to measure the isotopic ratios of the sample. U-Series Alpha-spectrometry dating was the first utilization of $^{238}$U decay, detecting and counting the ejected alpha particles. The counting statistics are on the order of a few precents of the $^{230}$Th/$^{238}$U ratio, resulting in a best-case $2\sigma$ internal errors of $\pm 10$ ka for an age ranging between 70 ka and 150 ka and more often than not $2\sigma$ internal errors closer to $\pm 20$ ka (Broecker and Thurber, 1965; Thurber et al., 1965). Therefore, the majority of early chronologies within the EAWIO have limited accuracy, and can only be generally assigned to one Marine Isotope Stage. It was not until the 1990s that mass-spectrometry, particularly thermal ionization mass spectrometry (TIMS, Edwards et al., 1987, 2003), began to bring down the $2\sigma$ error allowing MIS sub-stage discernibility. Additional advancements such as multi-collector inductively coupled plasma mass spectrometry (MC-ICPMS) have brought the $2\sigma$ uncertainties under ideal conditions down to $\pm 100$ a at 130 ka (Cheng et al., 2013). Besides biogenic carbonates, two terrestrial limiting chronologies from lithified dunes were established through the use of Luminescence (OSL). All uncertainties are stated at $2\sigma$ and, when needed, they have been converted from the $1\sigma$ values reported in the original papers.

 ## 2.4 Tectonics

The tectonic setting of PRSL indicators plays a significant role in their interpretation. Active faulting is found throughout the EAWIO (Figure 3a). For example, the majority of the East Africa coast is sitting atop the Somalia Plate that is slowly moving eastward as the East African Rift Zone (EARZ) slowly opens. Spreading rates in the EARZ decrease from north to south, 4.5 mm/a in Ethiopia to 1.5 mm/a along the Mozambique coastal plain (Stamps et al., 2008). While to the north, the Gulf of Aden

 is home to the Arabia-Danakil-Somalia triple junction.

When reported in literature, tectonic categories (stable, uplifting, or subsiding) are recorded within the database as to give the best possible picture of each sea level indicator setting. However, the magnitude of vertical land movements (VLM) are not explicitly included in the database. The reason behind this decision resides in the fact that the VLM rates that were reported in literature tend to be derived from several different assumed eustatic sea levels of the LIG. As this is directly tied to sea level, it

 therefore does not meet the strict "sea-level independent" criteria for insertion into the database.

## 2.5 Paleo Relative Sea Level Estimation

In order to extract paleo relative sea level (PRSL) from measured elevations, the IR and RWL for the measured indicator are needed (Shennan, 1982). The IR relies upon the measurement of modern upper and lower limits of the indicator in relation to an established datum. However, few studies have thoroughly documented the upper and lower limits of the site specific

 modern analogue to the indicator. To supplement missing IR and RWL values, Lorscheid and Rovere (2019) introduced a reliable empirical method that uses a global dataset of wave and tide model outputs in conjunction with the morpho- and hydrodynamic formation environment of the most common sea level indicators. This methodology was then packaged into a open-access stand-alone software, IMCalc, availible at: https://sourceforge.net/projects/imcalc/ (Lorscheid and Rovere, 2019). In the database we use the upper and lower limits when given by original authors, however, the majority of upper and lower

 limits for our PRSL points are calculated from the IMCalc software. While IMCalc provides reliable IR and RWL estimates, accurate surveying of local modern analogues is always preferable (e.g. Dutton et al., 2015). Once the upper and lower limits are determined, WALIS automatically calculates the IR, RWL, PRSL, and PRSL uncertainty, based on the schemes from Rovere et al. (2016a). All of the PRSL elevations in the following text have been calculated from originally published survey elevations using this methodology in order to standardize their comparison.

 ## 2.6 Uncertainties and Data Quality

The aim of WALIS is to provide the most objective evaluation of PRSL data as possible. It therefore must be explicitly noted that each data set is evaluated by a set of quality control standards that are used throughout the WALIS database (Table 4, Rovere et al., 2020). For the most part, elevation measurements were stated in plain language by the original authors, without describing in detail neither measurement methodologies nor measurement errors. We have therefore applied our best estimate

 errors in these cases based on the standard accuracy of the survey methodologies employed by the original authors (Table 2). When we have done so, we mention this in our evaluation of the RSL Proxy Quality inside the database.

As discussed previously (Section 2.3), $^{238}$Th/U ages are reliant upon the technique and transparency of metadata. While many earlier studies briefly refer to the methodology used, they often provide little, if any, analytical metadata. Within the database, we have accepted all $^{238}$Th/U ages as reported by the original authors and have only reported recalculated ages from Chutcharavan and Dutton (2020) which utilize $^{234}$U and $^{238}$Th decay constants from Cheng et al. (2013). Each chronological constraint has been rated using the common guidelines provided in the WALIS documentation (Table 5). Additionally, we have reported open-system ages for samples that are derived from mollusks (e.g. *T.Gigga*), which are widely accepted as providing inconsistent $^{238}$Th/U age reliability (e.g. Ayling et al., 2017) and therefore have been assigned a Marine Isotopic Stage designation rather than an outright age. Data with quality higher than 4 (good) are from the most recent studies within this region and are those who have adopted more rigorous sample screening procedures and have access to the most recent advances in mass-spectrometry (e.g. MC-ICPMS).

## 3 Sea Level Data Points

In total, our database counts 58 sea-level indicators and 2 terrestrial limiting points (Figure 2). All sea-level indicators are used to gather an associated PRSL based on the IM associated with the landform. As the EAWIO is a wide geographic area, we describe our data points based on a country by country rationale. We start in the north with Somalia down to Mozambique in the south. We then move offshore to the islands along the Mozambique channel, including Madagascar. Finally, we review sea-level histories of the Seychelles, Mayotte, Mauritius, and other small minor islands. PRSL indicators are referenced to their respective WALIS RSL ID number as well as their chronological constraint, if available. An additional interactive map of the EAWIO data is available in the supplementary material.

### 3.1 Somalia

The knowledge of Somalian Pleistocene sea-level indicators is limited. Only two main regions, the Gulf of Aden Coast (Sanaag) and Mogadishu (Banaadir), have been reported in the literature.

#### 3.1.1 Sanaag

The northern coast of Somalia (along the Gulf of Aden) is dominated by uplift and the Guban coastal plain. Here, Brook et al. (1996) described a series of four uplifted marine terraces that are intersected by ephemeral stream-carved gorges, locally known as Toggas. Mapping of the area was conducted using aerial photographs in conjunction with a series of four transects using altimeter measurements from the field (Brook et al., 1996). Coordinates of samples and terraces in the database were estimated in Google Earth from the original published maps.

Marine terraces are a relatively continuous feature along the northern coast of Somalia and are comprised of fluvial gravels mixed with marine sands that include a significant bioclast component (shell and coral debris). Brook et al. (1996) dated two surface samples with U-Series Alpha-spectrometry from a terrace at 8m above sea level (WALIS ID# 426). These two samples, entered in WALIS as BK96-003-001 & BK96-004-001 returned lower limit ages of 98 $\pm$ 8 ka and 99 $\pm$ 16 ka, respectively

(Table 6). Along one of the Togga walls, a larger *Favites* coral was sampled 3 m below the surface of the 8 m terrace (indicated as being 10 m above sea level in the original publication). This returned an age of $108 \pm 16$ ka, indicating that the 8 m terrace formed following the deposition of the coral, possibly within MIS 5c. Standing above the 8 m terrace, Brook et al. (1996) report that the 16 m terrace had fewer sampling opportunities. As a consequence, one surface sample is reported for this terrace (BK96-009-001, WALIS ID# 702), which was dated to $145 \pm 22$ ka, indicating a likely formation age of MIS 5e. The PRSLs for the 8 m and 16 m terrace are calculated at $+9.7 \pm 2.6$ and $+13.7 \pm 3.3$ m, respectively.

### 3.1.2 Banaadir

Moving south along the coast, Banaadir province is home to Somalia's capital and largest city: Mogadishu. The city as well as the surrounding area to the north and south is built upon Pleistocene reef deposits (Figure 4, Carbone and Accordi, 2000). Three RSL index points are described around the capital, one from Brook et al. (1996) and the other two from Carbone and Matteucci (1990); Carbone and Accordi (2000). Brook et al. (1996) describe a recently exposed quarry wall south of Mogadishu displaying a transgressive-regressive sequence of a coral and shell rich sandy layer sandwiched between beach sand deposits and then finally topped by a package of aeolian sand. While not in situ, a sample from this coral layer (BK96-010-001, WALIS ID# 703) was dated with by U-Series Alpha-spectrometry to $82 \pm 12$ ka. Brook et al. (1996) place the sample at 4 m above present sea level, we calculate the PRSL to be $+3.8 \pm 3.9$ m. Brook et al. (1996) connect this sample to the 2 m Terrace found along the northern coast of Somalia. Carbone and Accordi (2000) describe a Pleistocene reef terrace at, and to the south of Mogadishu standing 4 - 6 m high. They describe this terrace as characterized by a fringing reef complex composed of massive *Porites*, *Lobophyllia*, *Galaxea*, and *Acropora*. While no explicit survey is described, the originally reported 4 - 6 m elevation is used to calculate a PRSL of $+6.4 \pm 1.5$ m based on the description of the exposed quarry wall (WALIS ID #s 348 & 349, Carbone and Matteucci, 1990). Carbone and Accordi (2000) attribute this sequence to the Pleistocene based on unpublished ages between 105 and 131 ka (errors not stated). Approximately 65 km down the coast from Mogadishu, near the small city of Merka, Carbone and Accordi (2000) describe a sheltered, well-developed reef with massive corals in growth position. We calculated PRSL of $+6.4 \pm 1.5$ m and a correlated age to the terraces to the north of between 105 and 131 ka (WALIS ID# 351).

### 3.2 Kenya

The coastal region of Kenya can be divided into three sections: Northern, Central, and Southern (Oosterom, 1988). To the north, raised patch reefs are distributed amongst lagoonal, dune deposits, and beach ridges. Along the central coast, the patch-reefs transition into more developed fossil reef terraces. Finally to the south, solitary patch-reefs within lagoonal facies return. Oosterom (1988) tied the development of the central uplifted reefs to the topography of the hinterland and the lack of significant fluvial discharge, unlike to the north or south. That being said, marine limestone facies are nearly continuously exposed along the entire Kenyan coastline (Figure 5), with a maximum elevation of 15 m (Accordi et al., 2010). While there are several earlier studies focusing on the Pleistocene deposits along the Kenya coast (Battistini, 1969; Ase, 1981; Braithwaite, 1984), Accordi et al. (2010) are the first to provide vertical positioning data for samples taken. Additionally, Accordi et al. (2010) undertook

the most extensive dating of the emerged Kenyan reef sequence. Here, *Tridacna gigas* were sampled in situ and analyzed using U-Series Alpha-spectrometry. Due to variability of the calculated initial $^{234}U/^{238}U$ activity ratio of their samples (compared to the seawater value over the last 120 ka), Accordi et al. (2010) considered their samples as diagenetically altered, and used the open-system model of Scholz et al. (2004) to recalculate ages. Therefore, ages are treated with caution in WALIS, as mollusks 225 (i.e. *T. gigas*) are susceptible to inconsistent U-Series uptake and loss rates when compared to coral specimens (Ayling et al., 2017). The samples were split into three groups by Accordi et al. (2010) according to their apparent elevation and location along the coastline: Group A, Group B, Group C. Group A is scattered along the southern coast between the towns of Schimoni and Kalifi. The samples were taken from on top of the coral reef terrace within an elevation range of 8 - 15 m above mean sea level and have an open-system age of $120 \pm 8$ ka and a PRSL of between +14.5 m and + 21.5 m (MIS-5e, WALIS ID #s 189-192; 230 Table 6). Groups B and C are taken from the face of the limestone cliffs 0 and 6 m above mean sea level. Group B samples come from the central to northern section of coast between Kalifi and Manda Island, and have an open-system age of $118 \pm 14$ ka and a claculated PRSL of between +9.5 m and + 14.5 m (MIS-5e to 5d, WALIS ID#s 193-198; Table 6). Finally, Group C has an open-system age of $100 \pm 8$ ka and a claculated PRSL of between +8.5 m and +12.5 m (MIS-5c to 5d, WALIS ID#s 199-201) and is located along the same section of coast as Group A (Schimoni to Kalifi). Within the database, however, 235 we treat each sample independently in order to seperate interpretation from the raw data. Finally, calculated PRSL uses the indicative range described by Accordi et al. (2010) for *T.gigas* along the modern Kenyan coast, 3 to 10 m below MSL. In many instances there are poorly preserved massive or branching corals in growth position within this same facies.

### 3.3 Tanzania

Similar to the littoral of Kenya, Tanzania has nearly continuous marine limestone cliffs cropping out along the coast. Mainly 240 interpreted as fossil back-reef facies, the marine limestones are rich in *Halimeda* sediment and solitary coral mounds (Arthurton et al., 1999). From north to south, Tanzania's littoral zone can be divided into three regions: Tanga, Zanzibar Archipelago, and Dar-es-Salaam.

#### 3.3.1 Tanga

The northernmost coastal province of Tanga is characterized by a band of 8 m high emergent fringing reefs, a similar coastal 245 morphology as found in southern Kenya. Termed the "Azanian Series" by Stockley (1928), these emergent fringing reefs have been given a Pleistocene age and are abundant in large coral specimens as well as sponges and echinoids. Unfortunately, like those found in Kenya, the majority of specimens are re-crystallized, making U-Series dating particularly difficult (Cooke, 1974). We therefore do not include any specific PRSL proxies from the Tanga coastline within the database.

#### 3.3.2 Zanzibar Archipelago

The Zanzibar Archipelago comprise two main islands, Pemba and Unguja, along with several smaller islands stretching from the north of Tanzania down south to Dar-es-Salaam. The two main islands are generally constructed from extensive deposits of

the Azanian Series, forming two distinct terraces. The Older Azanian Limestone is a bivalve-gastropod packstone/ grainstone, indicative of a shallow low energy lagoon setting, that is presently found along the western fringe of Zanzibar (Stockley, 1928). This unit is of unknown age, but is described in the literature as being analogous to the 30 m "back-reef/lagoonal" facies along the Kenyan coast (Abuodha, 2004). Between the higher Older Azanian Limestone, and the lower Younger Azanian Limestone is a distinct erosional disconformity, indicating regression. The Younger Azanian Limestone is characterized by abundant coral mounds and bioherms, in growth position, surrounded by coral fragment grainstone. Several studies (e.g. Arthurton et al., 1999; Kourampas et al., 2015) have correlated the terrace formed from the Younger Azanian Limestone to the lower terrace along the coast of Kenya of MIS 5 age (Braithwaite, 1984). Arthurton et al. (1999) argues that the erosional surface of the marine terrace is of late-MIS 5 age because necessary erosion rates for the terrace to be of Holocene age far outpace the observed modern rates on Zanzibar as well as the lack of geological evidence for rapid sea-cliff retreat (i.e. talus deposits). Kourampas et al. (2015) provide a descriptive transect of the Jambiani marine terrace from which we calculate a PRSL of +11 $\pm$ 5.1 m for MIS 5 (WALIS ID# 212). Unfortunately, no current accurate ages are available and we can only postulate that the lower of the two terraces is correlated to those found in Kenya of MIS 5 age (Battistini et al., 1976).

### 3.3.3 Dar-es-Salaam

Named after the most populous city in Tanzania, the Dar-es-Salaam region sits facing the Zanzibar Archipelago. Here, the terraces from Tanga continue. Battistini (1966) provides the most recent description of the Pleistocene sections cropping out along the coast (Figure 6). At Ras Kankadya, Battistini (1966) describes a coral reef terrace with massive corals in growth position 6 - 7 m above high tide disconformably topped by a rubified aeolianite. From this the coral reef terrace we calculate a PRSL of +7.9 $\pm$1.1 m for (WALIS ID# 724). Chronologically, Battistini (1966) identifies this section as Reef II, which he attributes to the Karimbolian Limestone (MIS 5; Table 6) that he observed in northern Madagascar (Battistini, 1965b).

Reuter et al. (2010) describe a transgressive wetland sequence along the southern Tanzanian coast, near the town of Lindi. Siting at 21 m a MSL, the authors identified this deposit, using $^{14}$C dating of an *Assiminea* shell, to 44 ka. They argue that this section of coast has undergone 80-110 m of uplift since the last glacial maximum. While situated near the East African Fracture Zone, this would require a 1.8 to 2.5 mm/a uplift rate, which is not seen in other studies of the surrounding area. Kourampas et al. (2015) carried out an investigation of neighboring Zanzibar Island and commented on this age, citing that the $^{14}$C date is right on the borderline of being reliable (the $2\sigma$ values are between 48 - 41 ka) and should be dated using a different methodology that is more appropriate to that age range. Kourampas et al. (2015) also made the observation that Zanzibar is relatively stable (0.1 - 0.2 mm/a) uplift based on observations of more recent speleothems. Unfortunately, this is still left up to debate as no reliable U-Series or other chronology is available from the Tanzanian coast and only chronostratigraphy relating outcrops to neighboring Kenyan deposits is available.

### 3.4 Mozambique

Unlike the coral reef terrace deposits further north, Mozambique's coast is dominated by one of the world's largest coastal dune systems (Botha et al., 2003). Studies reporting on paleo sea-level indicators from this coast have been limited until

relatively recently. In southern Mozambique, near the border with South Africa, two locations, Inhambane and Maputo, have been investigated and chronologically constrained with OSL dating. While not sea-level indicators in their own right, these two locations provide terrestrial limiting points as well as a chronological constraint for a tidal notch.

### 3.4.1  Inhambane

Bazaruto Island, the largest of the Bazaruto Archipleago, is comprised of active and inactive dunes migrating across an older,
Pleistocene weathered aeolianite core (Armitage et al., 2006). At Zengueleme, on the eastern, bay-facing coast, an exposed $\tilde{2}0$ m tall bluff was described by Armitage et al. (2006). The base of the foramtion is composed of reddish aeolianite dated using OSL to 126 ± 24 ka (AR06-003-001, associated with WALIS ID# 184; Table 6). There is no PRSL for this formation as this is only a terrestrial limiting point. This aeolianite is then covered by a significantly younger dune sequence (23.8 ± 4.8 ka Armitage et al., 2006).

### 3.4.2  Maputo

Further south, near the border with South Africa, the Maputo region is home to the capital of Mozambique, Maputo. Just offshore the capital lies the island of Inhaca. An initial survey of Inhaca by Hobday (1977) describes a notch standing 5 - 6 m above modern sea level at the northernmost point of the island, Cabo Inhaca. This notch is then referred to again by Armitage et al. (2006) who obtained OSL dates for the aeolianite formation the notch is carved into, dating them to 150 ± 24 ka (AR06-
001-001, WALIS ID# 182; Table 6). This gives a maximum age constraint to the notch, and is therefore inferred to be of MIS 5 age. However, Armitage et al. (2006) indicate issues with the reliability of the OSL age, suggesting that this is possibly an underestimation of the age of the aeolianite sedimentation. This is highlighted by the $2\sigma$ of ± 24 ka. It should also be noted that Hobday (1977) provides extensive sedimentological descriptions of the calcarenite and dune facies found on Inhaca as well as mentions marine terraces between 5 - 6 m around the island, corresponding to the tidal notch at Cabo Inhaca. Unfortunately,
Hobday (1977) does not give specific locations of these terraces and are therefore not included in the database.

## 3.5  Madagascar

As the fourth largest island in the world and with over 4,800 km of coastline in tropical waters and limited terrestrial discharge, Madagascar provides excellent growth conditions for coral reefs. While not documented around the entire island, emergent reefs of Pleistocene age have been described, surveyed, and dated in two main regions of the island: the North and the South.

### 3.5.1  North

Situated at the northern tip of Madagascar, Antsiranana (formerly known as Diego-Suarez) and the surrounding coastline has been the subject of the most recent published PRSL record on the island. Stephenson et al. (2019) revisited sites previously described by Guilcher (1954) and Battistini (1965b). Battistini (1965a) first described two levels of emergent reefs at Cap d'Ambre, one ("Reef I") at 25 m elevation, which he infers to be older, and "Reef II" at 5 - 6 m elevation. Unfortunately, no

dating was carried out but, according to Battistini (1965b) "Reef I" is assumed to be Tatsimian and "Reef II" associated with the Karimbolian transgression. Just to the west of Antsiranana, on the Orangea peninsula, Battistini (1965a) described "Reef I" at 3 - 4 m and "Reef II" at 16 m. While no date at these two sites is available from this first expedition, Battistini et al. (1976) provides a U-Series Alpha-spectrometry age for the nearby Baie des Dunes of between $130 \pm 40$ ka and $160 \pm 30$ ka at 2 m above MSL. Discrepancies in the elevation between the sites were briefly discussed by Battistini (1965b) with possible explanations including active faulting, tilting, and volcanic subsidence due to the proximity of Mount Ambre.

Stephenson et al. (2019) revisited the Cap d'Ambre and Antsiranana coastlines. Four RSL proxies are extracted from the 13 available open-system U-Series dates and DGPS elevations obtained by Stephenson et al. (2019). At Cap d'Ambre, a coral reef terrace stands at $+9.3 \pm 1.2$ m MLWS with an age between $121.8 \pm 1.5$ and $125.5 \pm 1.8$ ka (ST18-002-001 and ST18-001-001, WALIS ID# 149, Table 6). This translates to a PRSL of $+10.7 \pm 1.4$ m MSL. Moving south along the eastern shoreline, near Baie des Dunes from Battistini et al. (1976), a coral reef terrace elevation from Cap Miné is recorded at $+6.8 \pm 1.2$ m above MLWS with an age between $125.5 \pm 1.8$ and $136.9 \pm 2.3$ ka (ST18-003-001 and ST18-004-001, WALIS ID# 159, Table 6). We calculate a PRSL of $8.2 \pm 1.4$ m from Cap Miné. These ages mirror the younger age of Baie des Dunes ($130 \pm 40$ ka) but are significantly younger than the older $160 \pm 30$ ka age from Battistini et al. (1976). The elevation reported by Stephenson et al. (2019) is also higher than that described by Battistini et al. (1976), 6.8 m vs. 2 m respectively. As with other datasets, the differences in analytical capability between the 1970s and late 2010s can certainly be a leading cause of these discrepancies in age and in elevation.

Approximately 25 km south of Cap Miné , Stephenson et al. (2019) observed the coral reef terrace again at Ankirikiriky Bay. Here, the terrace sits at $+4.3 \pm 1.2$ m above MLWS with an age of $129.4 \pm 1.8$ ka (ST18-005-001, WALIS ID# 161). This translates to a PRSL of $+ 5.3 \pm 1.5$ m. Finally, further south on the shore of Irodo Bay, the coral reef terrace is observed again, this time at an elevation of $+2.8 \pm 1.2$ m MLWS and an age of between $126.6 \pm 2$ and $141.8 \pm 1.9$ ka (ST18-005-001 to ST18-009-001, WALIS ID# 162; Table 6). This correlates to a PRSL of $+4.1 \pm 1.4$ MSL. The variance in RSL elevations is attributed to possible mantle convective upwelling, creating a dynamic topographic signature (Stephenson et al., 2019).

### 3.5.2 South

To the south of the coastal town of Tulear lies arid Madagascar spiny forestland dominated by thickets of *Euphorbia stenoclada* and *Alluaudia procera*. These thickets back the rocky shoreline where Battistini (1964) describes emergent reef sections near the small fishing village of Lembetabe. Unfortunately, this first publication does not provide enough metadata to derive a PRSL proxy. Luckily, however, Battistini et al. (1976) briefly describes the outcrop and provides the first U-Series Alpha-spectrometry age for this region. The emerged reef, with large in-situ corals in growth position, was described at 1 - 2 m above MSL with an age of $85 \pm 10$ ka (WALIS ID# 949, BA76-001-001). We have calculated a PRSL of $+3.3 \pm 1.7$ m. This succession is topped by the much younger Lavanonien aeolianite that contains continental mollusk shells and fragments of Elephant bird shell. Battistini et al. (1976) makes the observation that this much younger age than other emerged reefs of the Indian Ocean (Veeh, 1966; Montaggioni and Hoang, 1988) may be due to the exposed sections of reef being built in multiple phases.

The reef and sedimentary sequences at Lembetabe are the subject of an ongoing investigation led by the co-authors of this paper, which will include precise measurements, interpretations and MC-ICPMS U-Series ages. The results will be inserted in WALIS as soon as they will become available.

### 3.6 Seychelles

The Seychelles are the largest group of islands in the western Indian Ocean and represent a stable, far-field study site (Dutton et al., 2015). The main islands themselves are characterized by a granitic core with fringing reefs accreting in the subtidal zone to the bare rock. Outlying islands extend south and to the west, with the Aldabra Atoll representing the most westward extent.

### 3.6.1 Main Islands

The third-largest island of the granitic Seychelles archipelago, La Digue, has been the subject of several geomorphological investigations (e.g. Montaggioni and Hoang, 1988; Israelson and Wohlfarth, 1999; Wallinga and Cunningham, 2015). Dutton et al. (2015) describe coral colonies attached to granitic bedrock, similar to those observed within the present subtidal zone. From this, a total of five RSL indicators are included in the database. These are accompanied by 25 U-Series ages determined using MC-ICPMS (Dutton et al., 2015). We have included recalculated ages from Chutcharavan and Dutton (2020) in the database, however because these are the only recalculated ages within our database, we refer to the originally reported ages from Dutton et al. (2015) hereafter. Each sample was sub-sampled in triplicate and ages have been variance-weighted averaged from the sub-samples (Dutton et al., 2015). All PRSL elevations for Dutton et al. (2015) are determined using the modern analog scheme described by Vyverberg et al. (2018). At Inland (WALIS ID# 570, Table 6), indicator elevation was measured at +6.7 ± 0.2 m above MLWS and a PRSL was calculated to be +7.7 ± 1.0 m. Several in situ corals dated at this locality correlate this PRSL to MIS 5e, averaging 127.3 ± 0.9 ka. Moving out to the coast, the outcrop at Anse Source d'Argent has two subsites (Sites #7a and #8 in Dutton et al., 2015). Here, Site #7a (WALIS ID# 572) represents a re-sampling of the earlier Israelson and Wohlfarth (1999) mission to the island. This site sits at 7.4 ± 0.2 m above MLWS and the calculated PRSL is +8.4 ± 0.2 m. Accepted coral ages from this site average 127.3 ± 0.4 ka. Near Site #7a, Site #8 (WALIS ID# 573) has an elevation of +3.27 ± 0.2 m above MLWS and a PRSL +4.3 ± 1.0 m above MLWS. Site #8 has an average age of 127.1 ± 0.5 ka. On the southeastern coast of La Digue is Grande Anse (Site #11 in Dutton et al., 2015), in situ corals were surveyed +8.14 ± 0.2 m above MLWS and a PRSL was calculated at +9.1 ± 1.0 m above MLWS at 124.1 ± 0.5 ka (WALIS ID# 574, Table 6). Finally, to the northwest lies the smaller Curieuse Island where Dutton et al. (2015) describe in situ corals similar to the ones found on La Digue at the Turtle Pond (Site #19a in Dutton et al., 2015). Corals from this outcrop were dated average 125.2 ± 0.6 ka at an elevation of +6.6 ± 0.2 m above MLWS, equating to a PRSL +7.6 ± 1.0 m (WALIS ID# 575).

### 3.6.2 Outlying Islands

Closer to the east African mainland than the granitic Seychelles Archipelago sit the small nearly uninhabited islands of the Aldabra group. The Aldabra group is made up of four islands: Aldabra, Assumption, Astove, and Cosmoledo. Of these islands,

three have permanent settlements (one military base and one scientific research station). Aldabra, the largest atoll of the group
and namesake, is constructed predominately from emerged reef terraces (Braithwaite et al., 1973). Two sequential papers,
Thomson and Walton (1972); Braithwaite et al. (1973), provide a detailed morphological, sedimentological, and chronological
survey of the Aldabra Atoll. Two terraces, one at 8 m and another at 4 m above MSL were described (Braithwaite et al., 1973).
These terraces comprise of well preserved, often fairly large corals in growth position (Figure 7b). In-situ samples from the
upper coral terrace were collected and returned an average U-Series Alpha-spectrometry age of 127 ± 18 ka (TH72-001-001 to
TH72-008-001, WALIS ID# 591; Table 6) (Thomson and Walton, 1972). We calculate a PRSL of +8.5 ± 1.1 m for this upper
coral terrace.

Laying 30 km to the south of the main island of the Aldabra Atoll, Assumption island was the subject of a more recent survey
by Korotky et al. (1992). On Assumption Island, three marine terraces are described; 2-3 m, 4-8 m, and 10-14 m above MSL.
Within this general morphological context, Korotky et al. (1992) provide a detailed stratigraphic description with corresponding
U-Series ages (unknown if Alpha- or Mass-Spectrometry) from 4-6 m high exposed section of the "Marine Terrace III" on the
southern coast of the island (Figure 7a, c). At the base of this outcrop, a 1 m thick section of lagoonal cross-bedded calcarenite
is present and was dated to 127 ± 5.4 ka (KO92-002-001, WALIS ID# 733; Table 6). Lying above this is a 1.9 m thick section
of coral reef with in-situ (possibly in growth position?) massive corals dated to 115 ± 3.6 ka (KO92-001-001, WALIS ID#
734; Table 6). The reef section is disconformably terminated and is topped by a 1.5 m thick cross-bedded section of calcarenite
with isolated rounded pebbles and cobbles, dated to 96 ± 6 ka (KO92-003-001, WALIS ID# 731; Table 6). The whole section
is capped by a thin remnant layer of aeolianite of unknown age. From this section, three RSL index points can be extracted.
The oldest, the lagoonal deposit has a maximum age of 127 ± 5.4 ka and a PRSL estimation of 2.5 ± 2.3 m. Next, the reef
deposit represents a PRSL of +6.6 ± 2.3 m at 115 ± 3.6 ka. Finally, the tidal marsh deposit indicates a PRSL of +6.0 ± 3.0 m
with a minimum age of 96 ± 6 ka.

## 3.7 Mauritius

Dominating the central Indian Ocean, the island nation of Mauritius consists of two main island groups: the Chagos and the
Mascarene Archipelagos. Here, Battistini et al. (1976); Montaggioni (1972, 1976) describe emergent reefs along the majority
of the Mauritius coastline as well on the two small islets to the north of the main island: Île Plate and Îlot Gabriel. The reefs
are generally characterized as a framestone with large in-situ *Acropora*, *Pocilloporidae*, and *Faviidae* corals (Figure 8). The
morphological description of the reefs was accompanied by one U-Series Alpha-Spectrometry age from Veeh (1966). This
index stands between +1.5 and +2 m MSL, representing a PRSL of 3.1 ± 2.3 m at 110 ± 40 ka (VE66-012-001, WALIS ID#
427; Table 6). A second reef sequence is situated higher, between +5 to +6 m MSL, however, this section is not dated and
therefore not included in the database.

On the main island of Maruitius, Battistini et al. (1976) describes emergent fringing reefs with corals in growth position
at around 10 m above MSL. This indicator is difficult to locate as the position is described as in the village of Tamarin as
well as the "Rock of the Virgin Mary" in the village of Choisy along the southwestern coast of Maruitius. In the database, we
use the best estimated position for the "Rock of the Virgin Mary" as the indicator's position. Using IMCalc, we calculated a

PRSL of 11.7 ± 1.7 m. This is related to one U-Series Alpha-Spectrometry age from (Battistini et al., 1976) of 120 ± 40 ka (BA76-003-001, WALIS ID # 3638; Table 6).

## 3.8    Islands in the Mozambique Channel

In the Mozambique Channel, which separates Madagascar from mainland Africa, lies the small volcanic Comoros archipelago (Comoros and Mayotte) as well as four of the five French Scattered Islands (*Îles Éparses*): the Glorieuses, Juan de Nova, Bassas da India, and Europa. Detailed elevations and ages of Pleistocene stratigraphic sequences are only available for the Glorieuses islands, with limited reference to a Pleistocene-Holocene contact on Mayotte.

### 3.8.1    Glorieuses Islands

Sitting approximately 200 km west of the northern tip of Madagascar, the Glorieuses is made up of two main islands: Grand Glorieuse and Ile du Lys (Guillaume et al., 2013). Battistini et al. (1976) provided the first U-Series Alpha-Spectrometry ages for the island. On Grand Glorieuse, an emergent reef with corals in growth position was sampled and an age of 150 ± 40 ka (BA76-005-001, WALIS ID# 307; Table 6) was established. This reef was described at 3 m above high tide level (HTL), which translates to a PRSL of +4.5 ± 2.0 m HTL. To the west of the main island, Ile du Lys is significantly smaller but has the better preserved Pleistocene record of the the two islands. Here, Battistini et al. (1976) describes an emergent reef outcrop between +3 to +5 m HTL. U-Series age of 159 ± 40 ka was obtained for a sample of in-situ coral (BA76-006-001; Table 6).

Guillaume et al. (2013) returned to the islands and provided 19 new, U-Series Alpha-Spectrometry ages for the islands. From these ages, four RSL proxies were established. At Cap Vert on the central west coast of Grand Glorieuse, sampled corals have elevations between +3.8 ± 0.2 and +3.5 ± 0.2 m MLWS and ages between 123.3 ± 12.6 and 140 ± 8.2 ka (GU13-001 to GU13-004, WALIS ID# 164; Table 6). This equates to a PRSL of +5.1 ± 0.5 m.

Just off the southern tip of the island, at Rocher Sud, one coral sample was dated to 127 ± 4.6 ka (GU13-005-001, WALIS ID# 173; Table 6) from an outcrop +4.4 ± 0.2 m MLWS equating to a PRSL of +5.9 ± 0.5 m. GU13-006-001 was also sampled from the same outcrop at Rocher Sud, however the age of 137 ka was rejected by the authors because of high initial $^{234}$U/$^{238}$ ratios Guillaume et al. (2013). Similar to Rocher Sud, the slightly larger Rocher Vert sits roughly 3 km to the northeast of the island emerging from the modern reef flat. Two samples were taken from Rocher Vert (GU13-007-001 and GU13-008-001, WALIS ID# 174; Table 6), however both samples were rejected as overestimating age based on especially high initial $^{234}$U/$^{238}$ ratios. The maximum elevation of the reef facies from Rocher Vert were none the less used within the database and an PRSL of +6.3 ± 0.5 m was calculated.

Ile du Lys is home to the island group's most extensive elevated reef flat exposure. Unfortunately, only one sample, GU13-014-001, passed the calcite screening process of Guillaume et al. (2013) and has an age of 124 ± 6.4 ka (WALIS ID# 175, Table 6). Two different lithologies separated by a disconformity are described. The lower of the two facies transitions from a *Halimeda* floatstone to a framestone dominated by branching and a few massive corals in growth position. GU13-014-001 was sampled from this floatstone layer. Above this, a discontinuity separates the overlying bed of coarse *Halimeda* rich grainstone

with occasional coralline bioclasts interpreted as an overwash deposit. The upper elevation of the lower unit is used as a RSL proxy and results in a PRSL of +6.4± 0.7 m above MSL.

### 3.8.2   Mayotte

The island of Mayotte is characterized by a tall rugged volcanic core surrounded by an almost continuous barrier reef. Camoin et al. (1997) conducted a reef drilling campaign to ascertain the growth patters of the reef during the Holocene. At the base of of the Holocene core section, -16 to -20 m below reef surface, a basal contact with what is believed to be a Pleistocene reef sequence was found. Unfortunately, samples from this section of the core had undergone significant diagenisis and no definitive age was found (Camoin et al., 1997). Due to the lack chronological constraints, this is not included in the database.

## 4   Further Details

### 4.1   Last Interglacial Sea Level Fluctuations

Sea-level fluctuations during the LIG, subsequent rises and falls within MIS 5, have been alluded to by several studies in the EAWIO region. For example Montaggioni and Hoang (1988), argue for two peaks, one between 139-133 ka and another at about 123 ka based on the distribution of their U-Series Alpha-Spectrometry ages across the granitic Seychelles. Brook et al. (1996) also identify apparent fluctuations in LIG sea level. Both the 8 m terrace and 16 m terrace they identified along the northern coast of Somalia (Section 3.1.1) are both most likely from the LIG. Here, there is stratigraphic evidence that regression occurred following the formation of the 16 m terrace before the 8 m terrace incised this alluvial unit. However, the magnitude of this fluctuation is overshadowed by two caveats: this coastal region is tectonically active and the 16 m terrace age is base on one sample (BK96-009-001, WALIS ID# 702) that Brook et al. (1996) call, "extremely questionable."

It has not been until recently that surveying methodology and chronological constraints have achieved an accuracy that enables the documentation of such fluctuations (Figure 3a & b). Vyverberg et al. (2018) conducted a multidisciplinary investigation of the Seychelles record of Dutton et al. (2015). Across multiple outcrops around the main islands, reef growth is interrupted by discontinuities within the paleo record. Vyverberg et al. (2018) argue that this interruption in coral growth is the possible result of subaerial exposure during a fall in sea level or a still stand. Braithwaite (2020) revisited Braithwaite et al. (1973) and describes evidence of variations in sea level during the LIG on Aldabra. However, both studies conclude that higher resolution dating is needed in order to confirm this hypothesis.

### 4.2   Other Interglacials

Within the EAWIO basin, older, less preserved marine deposits have been described in the literature. Unfortunately, the majority of these deposits have not been confidently dated. The only dated deposit is found on the central Kenyan coast near South Kilifi where Battistini et al. (1976) describes an emerged reef 2 m above HTL with a U-Series age of 240 ± 80 ka.

## 5 Future Research Directions

The recent studies of Dutton et al. (2015), Vyverberg et al. (2018) and Stephenson et al. (2019) show the potential for high-resolution data acquisition within the EAWIO (Figure 3b). Many earlier studies have described localities that are promising for LIG sea-level specific studies, future studies should revisit these sites with modern surveying and chronological methodologies to achieve more accurate constraints. For example, long stretches of raised coral reef terraces along the east African coast extending south from Mogadishu have the potential to provide an uninterrupted sequence of reef stratigraphy across hundreds

of kilometers. Many other of the coralline islands have been described by original geographic surveys and have not been revisited to properly survey or date elevated reef deposits (e.g. Astove Island (Bayne et al., 1970a, b) and Cosmoledo Atoll (Bayne et al., 1970a)) and have the potential to provide additional high-resolution RSL proxies.

## 6 Data availability

The East Africa and the Western Indian Ocean database is available at: https://doi.org/10.5281/zenodo.4043366 (Version 1.03)
(Boyden et al., 2020). The description of the database fields can be found at:https://doi.org/10.5281/zenodo.3961543 (Rovere et al., 2020).

*Author contributions.* PB compiled the database with extensive help from JWA on translating older French publications into English. AR is the main developer of WALIS. PB wrote the initial manuscript, with significant input from JWA and AR. Further input on the manuscript was provided by PD and DO.

*Competing interests.* The authors declare no competing interests.

*Acknowledgements.* The data used in this study were compiled in WALIS, a sea-level database interface developed by the ERC Starting Grant "WARMCOASTS" (ERC-StG-802414), in collaboration with PALSEA (PAGES / INQUA) working group. The database structure was designed by A. Rovere, D. Ryan, T. Lorscheid, A. Dutton, P. Chutcharavan, D. Brill, N. Jankowski, D. Mueller, M. Bartz, E. Gowan and K. Cohen. The data points used in this study were contributed to WALIS by Patrick Boyden, with P.Chutcharavan providing assistance with some U-Series data entry. Support for Patrick Boyden was provided by the DFG (Deutsche Forschungsgemeinschaft) Excellence Cluster "EXC 2077: The Ocean Floor – Earth's Uncharted Interface" (Project number: 390741603). Jennifer Weil-Accardo post-doctoral fellowship was funded by the French National Institute for Sustainable Development (IRD).

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

| Age | Malagasy Quaternary | Transgressions | MIS Stage |
|---|---|---|---|
| 6 | Upper Aepyornian | | |
| 14 | Middle Aepyornian | Flandrian | 3 |
| 82 | | Karimbolian | 5 |
| 130 | | | |
| 243 | Lower Aepyornian | Tatsimian | 7 |

**Figure 1.** Malagasy Quaternary nomenclature (Age is in ka, scheme modified after Battistini, 1984).

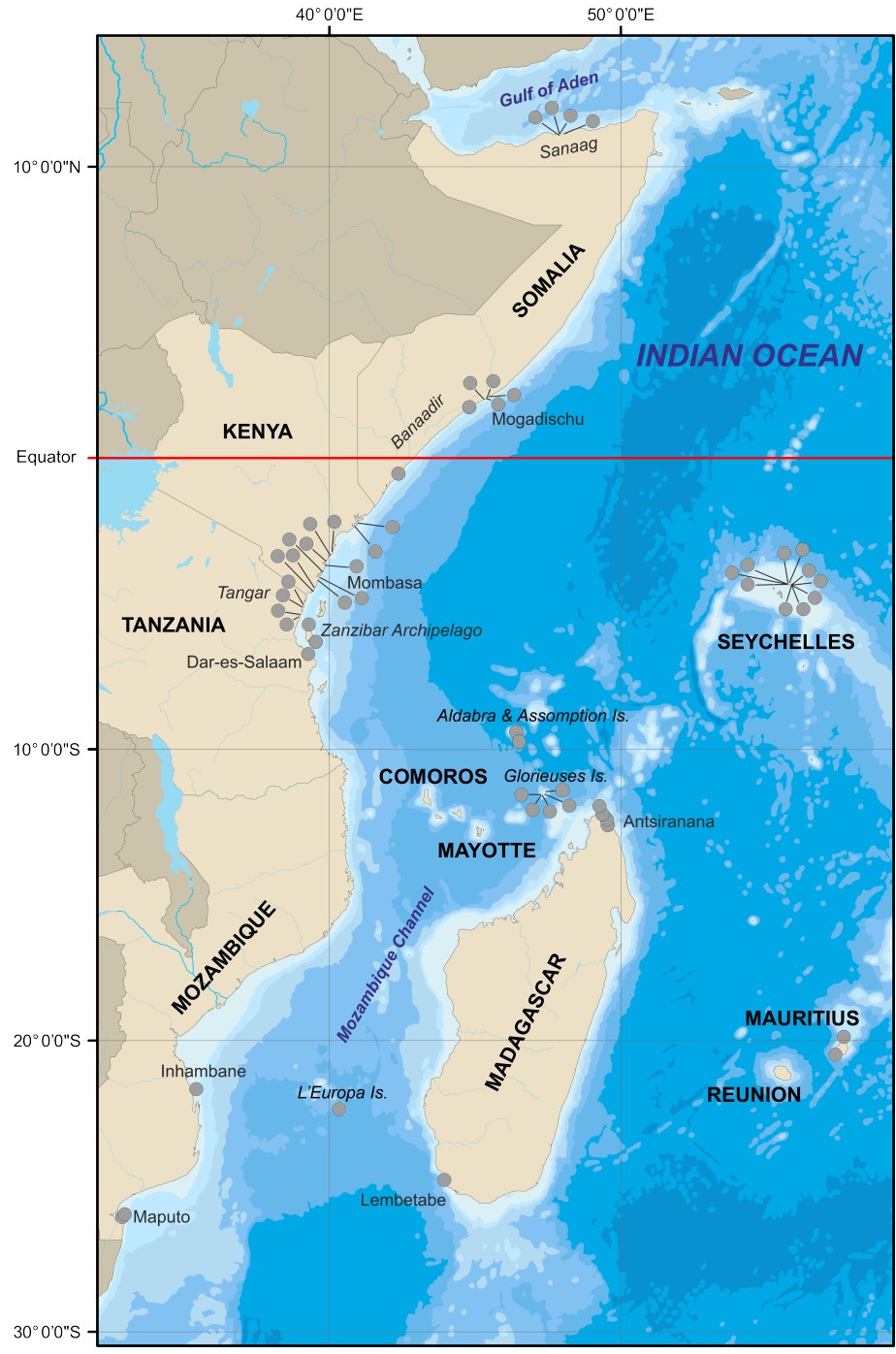

**Figure 2.** Overview map showing the distribution of PRSL proxies and their respective indicator types. Base map data compiled from Natural Earth; vector and raster map data @ naturalearthdata.com. An interactive map of the EAWIO data is available in the supplementary material.

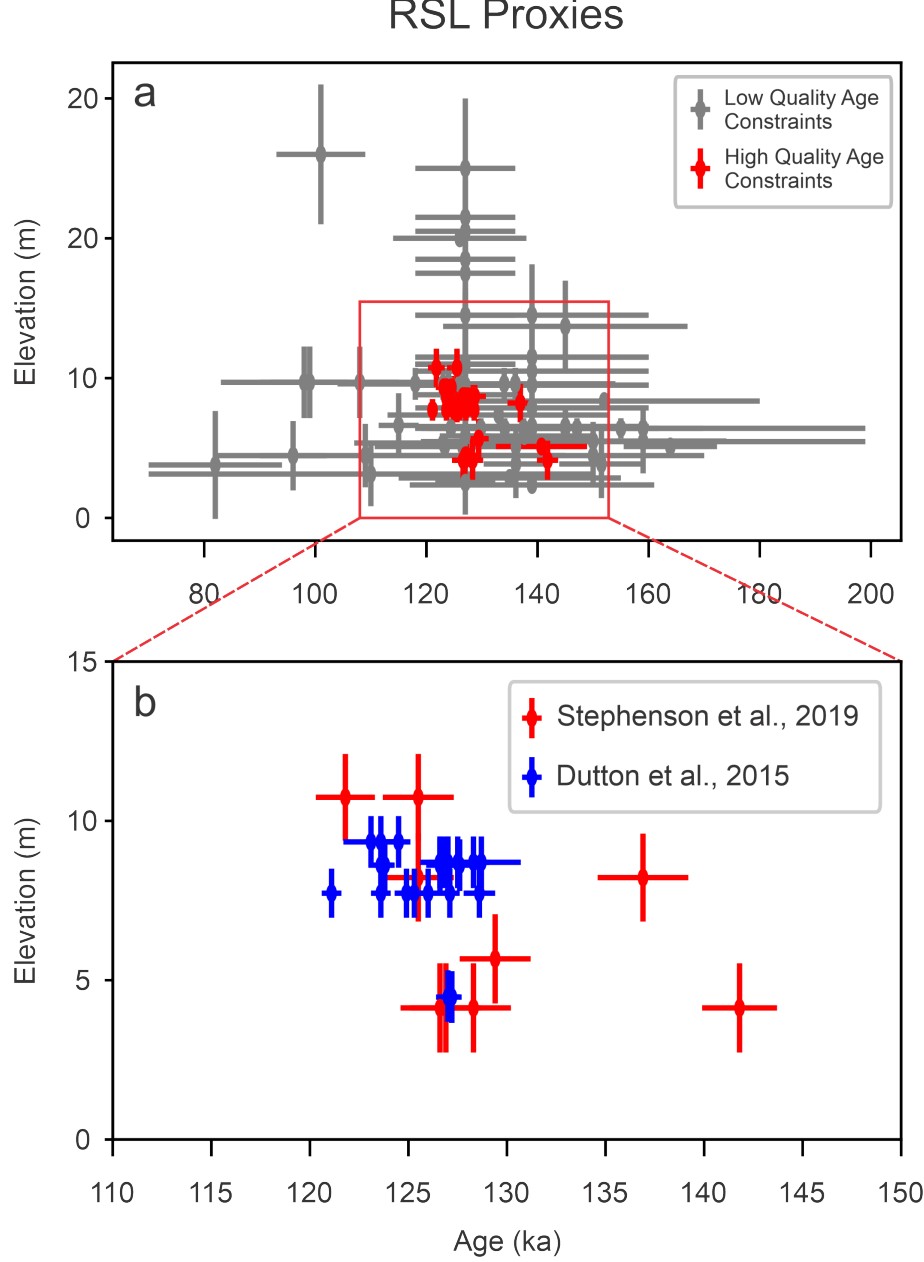

**Figure 3.** (a) Overview of age and elevation distribution of PRSL of the East Africa and Western Indian Ocean Region. Proxies with age constraints with a quality rating lower than 4 within WALIS are colored grey and those proxies with a quality rating of 4 or greater are colored red. The elevations refer to paleo relative sea level. "Relative" means that they are still uncorrected for any post-depositional vertical movement, such as, for example, tectonics or GIA. (b) The high quality age constraints are from two areas, Northern Madagascar (Stephenson et al., 2019) and the granitic Seychelles (Dutton et al., 2015).

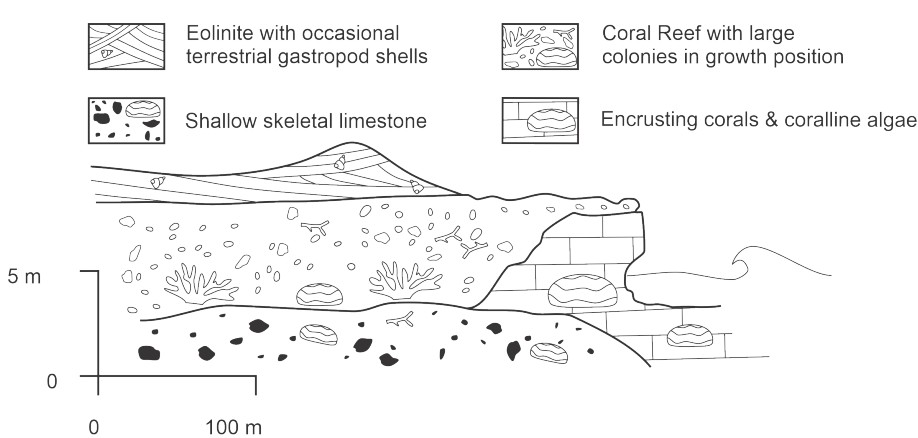

**Figure 4.** Raised reef and aeolinites of southern Somalia (Modified after Carbone and Accordi, 2000).

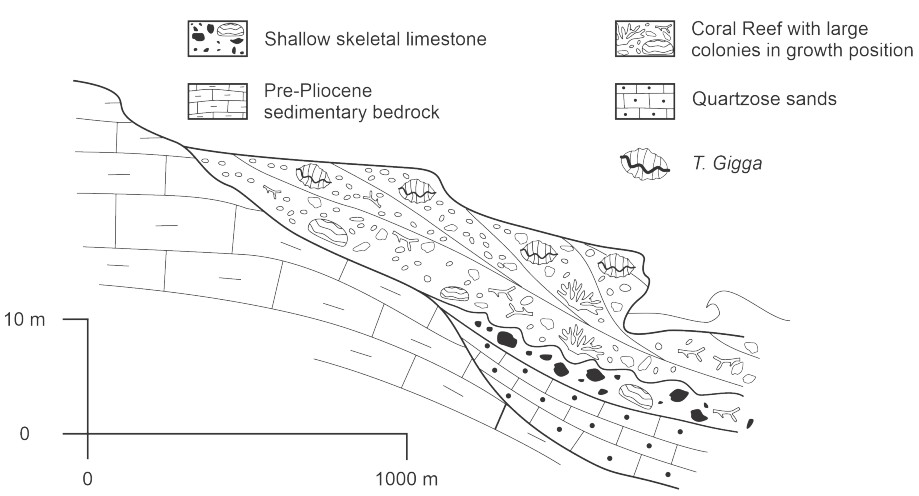

**Figure 5.** Litholog of emerged reef coast of Kenya (Modified after Accordi et al., 2010).

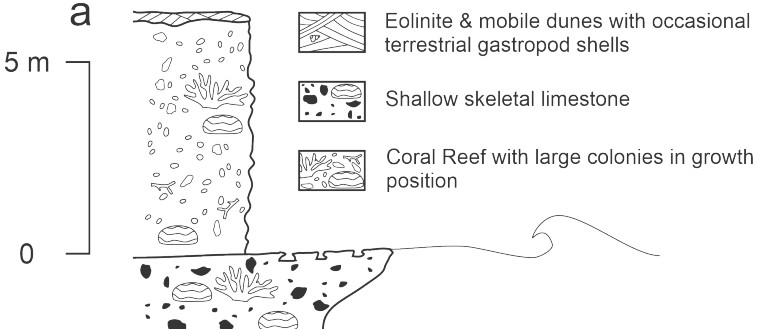
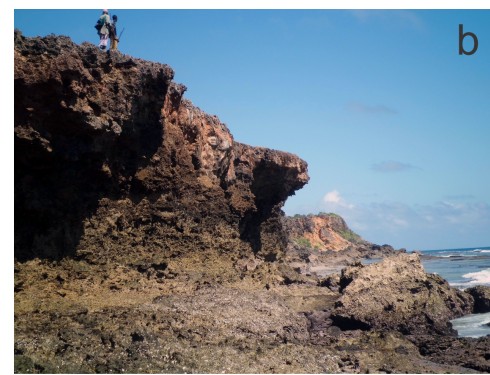

**Figure 6.** (a) Litholog of emerged reef near Dar-es-Salaam (Modified after Battistini, 1966). (b) Ras Mwanamkuru just south of Ras Kankadya. Exposed Pleistocene reef sitting atop poorly consolidated skeletal limestone. Approximately 10 - 15 m tall. Photo by D.Oppo.

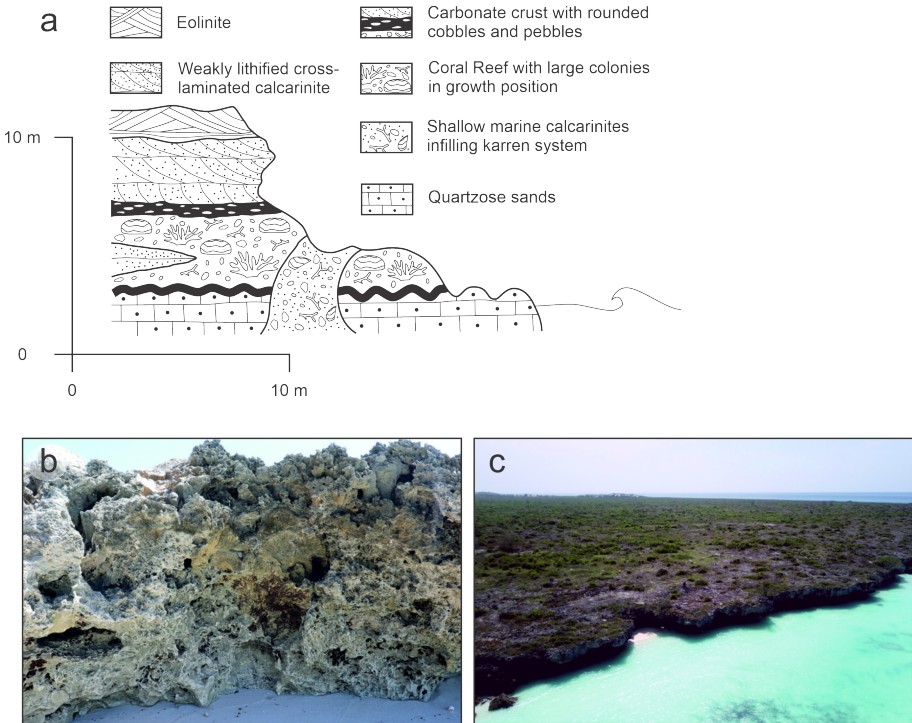

**Figure 7.** (a) Litholog of Marine Terrace II (Modified after Korotky et al., 1992). (b) Lower part of the Pleistocene reef complex at Aldabra Island. (c) Aerial view of the Pleistocene reef at Assomption Island. Photos by A. Rovere.

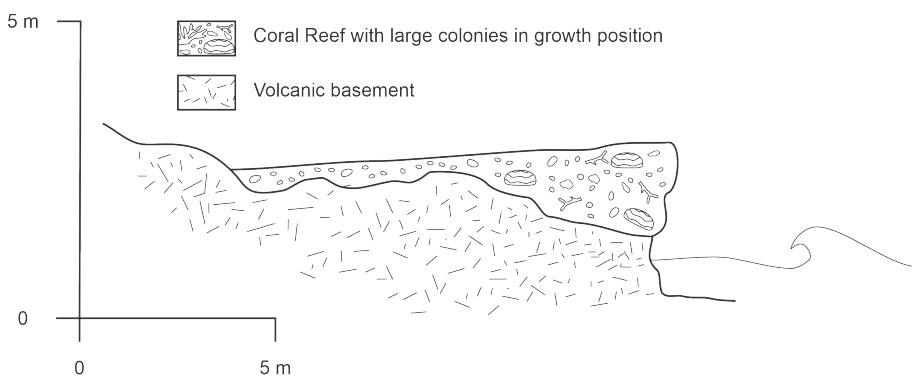

**Figure 8.** Raised reef platorm of Île Plate (Modified after Montaggioni, 1982).

**Table 1.** Relative sea-level indicators and their indicative ranges. Exported from Boyden et al. (2020)

| Name of RSL indicator | Description of RSL indicator | $RWL_{descr}$ | Description of IR |
|---|---|---|---|
| Coral reef terrace (general definition) | Coral-built flat surface, corresponding to shallow-water reef terrace to reef crest. The definition of indicative meaning is derived from Rovere et al. (2016b), and it represents the broadest possible indicative range, that can be refined with information on coral living ranges. | (Mean Lower Low Water + Breaking depth)/2 | Mean Lower Low Water - Breaking depth |
| Lagoonal deposit | Lagoonal deposits consist of silty and clayey sediments, frequently characterized by the presence of brackish or marine water fauna Rovere et al. (2016b). Usually, lagoon sediments are horizontally laminated Zecchin et al. (2004). Definition of indicative meaning from Rovere et al. (2016b). | (Mean Lower Low Water + modern Lagoon depth)/2 | Mean Lower Low Water - modern Lagoon depth |
| Marine Terrace | From Pirazzoli (2005): "Any relatively flat surface of marine origin". Definition of indicative meaning from Rovere et al. (2016b). | (Storm wave swash height + Breaking depth) / 2 | Storm wave swash height - Breaking depth |
| Shallow water coral reef facies | For use when specific reference is given to a modern coral species and or morphological occurrence. | Mean Low Water Springs | Mean Low Water Springs to depth of modern living analogue. |
| Tidal inlet facies | Coarse-grained, thickly bedded, trough cross bedding, herringbone cross bedding, multiple scours, Ophiomorpha and Skolithos trace fossils. | -0.5 m MSL to -3.5 m MSL | -0.5 m MSL to -3.5 m MSL |
| Tidal notch | Tidal notches are "indentations or undercuttings cut into rocky coasts by processes acting in the tidal zone (such as tidal wetting and drying cycles, bioerosion, or mechanical action)" Antonioli et al. (2015). Definition of indicative meaning from Rovere et al. (2016b). | (Mean Higher High Water + Mean Lower Low Water)/2 | Mean Higher High Water - Mean Lower Low Water |
| The datapoint is a marine or terrestrial limiting indicator | See detailed indicator description | No RWL Available | No IR available |

**Table 2.** Measurement techniques used to survey the elevation of last interglacial shorelines in the EAWIO region. Exported from Boyden et al. (2020).

| Measurement technique | Description | Typical accuracy |
|---|---|---|
| Barometric altimeter | Difference in barometric pressure between a point of known elevation (often sea level) and a point of unknown elevation. Not accurate and used only rarely in sea-level studies | Up to ±20% of elevation measurement |
| Differential GPS | GPS positions acquired in the field and corrected either in real time or during post-processing with respect to the known position of a base station or a geostationary satellite system (e.g. Omnistar). Accuracy depends on satellite signal strength, distance from base station, and number of static positions acquired at the same location. | ±0.02/±0.08 m, depending on survey conditions and instruments used (e.g., single-band vs dual-band receivers) |
| Total station or Auto/hand level | Total stations or levels measure slope distances from the instrument to a particular point and triangulate relative to the XYZ coordinates of the base station. The accuracy of this process depends on how well defined the reference point and on the distance of the surveyed point from the base station. Thus, it is necessary to benchmark the reference station with a nearby tidal datum, or use a precisely (DGPS) known geodetic point. The accuracy of the elevation measurement is also inversely proportional to the distance between the instrument and the point being measured. | ±0.1/±0.2m for total station ±0.2/±0.4 m for hand level |
| Not reported | The elevation measurement technique was not reported, most probably hand level or metered tape. | 20% of the original elevation reported added in root mean square to the sea level datum error |

**Table 3.** Sea level datums reviewed in this study, modified and exported from Boyden et al. (2020).

| Datum name | Datum description | Datum uncertainty | Reference(s) |
|---|---|---|---|
| High Tide Level | Described by Kennedy et al (2007) as the swash limit and the extent of fixed biological indicators, such as molluscs, having a restricted vertical range. | Per Rees-Jones et al. (2000), accurate to $\pm$ 2 m up to 15 m a.h.s.l and $\pm$ 5-10 m above 15m a.h.s.l. Uncertainty will be dependent upon measurement method. | Kennedy et al. (2007); Rees-Jones et al. (2000) |
| Mean Low Water Springs | "The average of the heights ... of each pair of successive low waters during that period of about 24 hours in each semi-lunation (approximately every 14 days), when the range of the tide is greatest." | Declared $\pm$ 0.1 m if datum is derived from 1 year and $\pm$ 0.25 m if measured over 1 month. | Baker and Watkins (1991) |
| Mean Sea Level / General definition | General definition of MSL, with no indications on the datum to which it is referred to. | A datum uncertainty may be established on a case-by-case basis. | |
| Not reported | The sea level datum is not reported and impossible to derive from metadata. | N/A | |

**Table 4.** Quality rating guidelines used for evaluating PRSLs. Exported from WALIS (Rovere et al., 2020)

| Description | Quality Rating |
| --- | --- |
| Elevation precisely measured, referred to a clear datum and RSL indicator with a very narrow indicative range. Final RSL uncertainty is submetric. | 5 (excellent) |
| Elevation precisely measured, referred to a clear datum and RSL indicator with a narrow indicative range. Final RSL uncertainty is between one and two meters. | 4 (good) |
| Uncertainties in elevation, datum or indicative range sum up to a value between two and three meters. | 3 (average) |
| Final paleo RSL uncertainty is higher than three meters. | 2 (poor) |
| Elevation and / or indicative range must be regarded as very uncertain due to poor measurement / description / RSL indicator quality. | 1 (very poor) |
| There is not enough information to accept the record as a valid RSL indicator (e.g. marine or terrestial limiting). | 0 (rejected) |

**Table 5.** Quality rating guidelines used for evaluating age information. Exported from the WALIS (Rovere et al., 2020)

| Description | Quality Rating |
|---|---|
| Very narrow age range, e.g. few ka, that allow the attribution to a specific timing within a substage of MIS 5 (e.g. 117 $\pm$ 2 ka) | 5 (excellent) |
| Narrow age range, allowing the attribution to a specific substage of MIS 5 (e.g., MIS 5e). | 4 (good) |
| The RSL data point can be attributed only to a generic interglacial (e.g. MIS 5). | 3 (average) |
| Only partial information or minimum age constraints are available. | 2 (poor) |
| Different age constraints point to different interglacials. | 1 (very poor) |
| Not enough information to attribute the RSL data point to any pleistocene interglacial. | 0 (rejected) |

**Table 6.** Summary of RSL proxies and terrestrial limiting points included in the WALIS database. [1]CR Terrace - coral reef terrace (general definition) SW CR - shallow water coral reef facies, MT - marine terrace. [2]NR - Not Reported. [3]Quality ratings are on a scale of 5 (Excellent) to 0 (Rejected), based on criteria from Rovere et al. (2020)

| RSL ID | Site | Lat (°) | Lon (°) | RSL Type[1] | Datum | PRSL (m) | Age | RSL Quality[2] | Age Quality[2] |
|---|---|---|---|---|---|---|---|---|---|
| 149 | Cap d'Ambre | -11.95 | 49.27 | CR Terrace | MLWS | $10.7 \pm 1.4$ | U-Series | 4 | 5 |
| 159 | Cap Miné | -12.24 | 49.38 | CR Terrace | MLWS | $8.2 \pm 1.4$ | U-Series | 4 | 4 |
| 161 | Ankirikiriky Bay | -12.41 | 49.53 | CR Terrace | MLWS | $5.7 \pm 1.4$ | U-Series | 4 | 5 |
| 162 | Irodo | -12.61 | 49.56 | CR Terrace | MLWS | $4.1 \pm 1.4$ | U-Series | 4 | 5 |
| 164 | Glorieuses Is. - C. Vert | -11.59 | 47.29 | CR Terrace | MLWS | $5.1 \pm 0.5$ | U-Series | 4 | 4 |
| 173 | Glorieuses Is. - R. Sud | -11.59 | 47.30 | CR Terrace | MLWS | $5.9 \pm 0.5$ | U-Series | 4 | 3 |
| 174 | Glorieuses Is. - R. Vert. | -11.57 | 47.33 | CR Terrace | MLWS | $3.9 \pm 2.4$ | U-Series | 4 | 1 |
| 175 | Glorieuses Is. - I.d.Lys | -11.52 | 47.38 | SW CR | MLWS | $6.4 \pm 0.7$ | U-Series | 4 | 2 |
| 182 | Cabo Inhaca | -25.97 | 32.99 | Tidal notch | MSL | $5.5 \pm 1.4$ | Luminescence | 1 | 1 |
| 183 | Barreira Vermelha | -26.06 | 32.90 | Terrestrial | MSL | N/A | Luminescence | 0 | 3 |
| 184 | Zengueleme | -21.67 | 35.44 | Terrestrial | MSL | N/A | Luminescence | 0 | 3 |
| 187 | Kikambala Quarry | -3.92 | 39.78 | CR Terrace | MSL | $21.5 \pm 3.6$ | U-Series | 2 | 3 |
| 189 | Msambweni Quarry | -4.46 | 39.49 | CR Terrace | MSL | $20.5 \pm 3.6$ | U-Series | 2 | 3 |
| 190 | Black Cliff | -4.20 | 39.62 | CR Terrace | MSL | $17.5 \pm 3.6$ | U-Series | 2 | 3 |
| 191 | Shelly Beach Quarry | -4.10 | 39.67 | CR Terrace | MSL | $18.5 \pm 3.6$ | U-Series | 2 | 3 |
| 192 | Takaungu | -3.69 | 39.86 | CR Terrace | MSL | $14.5 \pm 3.6$ | U-Series | 2 | 3 |
| 193 | Watamu - C. Track | -3.30 | 40.10 | CR Terrace | MSL | $14.5 \pm 3.6$ | U-Series | 2 | 3 |
| 194 | Watamu - Sea Cliff | -3.36 | 40.04 | CR Terrace | MSL | $11.5 \pm 3.6$ | U-Series | 2 | 3 |
| 195 | Manda Island - North | -2.24 | 41.00 | CR Terrace | MSL | $11.5 \pm 3.6$ | U-Series | 2 | 3 |
| 196 | Manda Island - South | -2.33 | 40.92 | CR Terrace | MSL | $9.5 \pm 3.6$ | U-Series | 2 | 3 |
| 197 | Kilifi Quarry | -3.56 | 39.91 | CR Terrace | MSL | $10.5 \pm 3.6$ | U-Series | 2 | 3 |
| 198 | Ros Ngomeni | -2.99 | 40.24 | CR Terrace | MSL | $9.5 \pm 3.6$ | U-Series | 2 | 3 |
| 199 | Funzi Island | -4.59 | 39.45 | CR Terrace | MSL | $12.5 \pm 3.6$ | U-Series | 2 | 3 |
| 200 | Mwasaro Village | -4.61 | 39.40 | CR Terrace | MSL | $8.5 \pm 4.2$ | U-Series | 3 | 3 |
| 201 | Diani Beach | -4.29 | 39.60 | CR Terrace | MSL | $8.5 \pm 3.6$ | U-Series | 3 | 3 |
| 207 | Ras Nungwi | -5.72 | 39.30 | CR Terrace | HTL | $8.8 \pm 5.6$ | Chronostrat. | 1 | 1 |
| 208 | Kigombe | -5.30 | 39.06 | CR Terrace | HTL | $11.0 \pm 2.2$ | Chronostrat. | 1 | 1 |
| 209 | Mwamani Bay | -5.13 | 39.11 | CR Terrace | MSL | $9.5 \pm 5.4$ | Chronostrat. | 1 | 1 |
| 210 | Yambe Island | -5.11 | 39.16 | CR Terrace | MSL | $11.0 \pm 5.4$ | Chronostrat. | 1 | 1 |

*Continued on next page*

| RSL ID | Site | Lat (°) | Lon (°) | RSL Type[1] | Datum | PRSL (m) | Age | RSL Quality[2] | Age Quality[2] |
|---|---|---|---|---|---|---|---|---|---|
| 211 | Ulenge Island | -5.00 | 39.17 | CR Terrace | MSL | $11.0 \pm 5.4$ | Chronostrat. | 1 | 1 |
| 212 | Jambiani - Old | -6.32 | 39.54 | CR Terrace | MSL | $25.0 \pm 5.0$ | Chronostrat. | 1 | 1 |
| 213 | Jambiani - Young | -6.32 | 39.54 | CR Terrace | MSL | $11.0 \pm 5.1$ | Chronostrat. | 1 | 1 |
| 307 | Glorieuses Is. - Grand | -11.59 | 47.29 | CR Terrace | HTL | $4.5 \pm 2.0$ | U-Series | 2 | 3 |
| 308 | Iles d'Europa | -22.36 | 40.35 | CR Terrace | MSL | $4.5 \pm 2.3$ | Chronostrat. | 2 | 0 |
| 348 | Mogadishu - Airport | 2.01 | 45.31 | CR Terrace | MSL | $6.4 \pm 1.5$ | U-Series | 2 | 2 |
| 349 | Mogadishu - Refinery | 1.98 | 45.23 | CR Terrace | MSL | $6.4 \pm 1.5$ | U-Series | 2 | 2 |
| 350 | Fuma Island | -0.55 | 42.38 | CR Terrace | MSL | $6.5 \pm 1.5$ | U-Series | 2 | 2 |
| 351 | Merka | 1.74 | 44.81 | CR Terrace | MSL | $6.4 \pm 1.5$ | U-Series | 2 | 2 |
| 419 | Ras Kalwein - C | 11.11 | 47.98 | MT | MSL | $9.7 \pm 2.6$ | U-Series | 2 | 2 |
| 426 | Ras Kalwein - A | 11.12 | 47.90 | MT | MSL | $9.7 \pm 2.6$ | U-Series | 2 | 2 |
| 427 | Plate and Gabriel | -19.88 | 57.66 | CR Terrace | NR | $3.1 \pm 2.3$ | U-Series | 2 | 3 |
| 570 | Inland | -4.36 | 55.83 | SW CR | MLWS | $7.7 \pm 1.0$ | U-Series | 4 | 5 |
| 572 | Anse Source d'Argent | -4.37 | 55.83 | SW CR | MLWS | $8.4 \pm 1.0$ | U-Series | 4 | 5 |
| 573 | Anse Source d'Argent | -4.37 | 55.83 | SW CR | MLWS | $4.3 \pm 1.0$ | U-Series | 4 | 5 |
| 574 | Grande Anse | -4.38 | 55.83 | SW CR | MLWS | $9.1 \pm 1.0$ | U-Series | 4 | 5 |
| 575 | Turtle Pond | -4.28 | 55.76 | SW CR | MLWS | $7.6 \pm 1.0$ | U-Series | 4 | 5 |
| 591 | Aldabra | -9.42 | 46.42 | CR Terrace | MSL | $9.6 \pm 1.1$ | U-Series | 1 | 3 |
| 702 | Ras Kalwein - A | 11.11 | 47.90 | MT | MSL | $13.7 \pm 3.3$ | U-Series | 2 | 2 |
| 703 | Mogadishu | 1.99 | 45.25 | MT | MSL | $3.8 \pm 3.9$ | U-Series | 2 | 2 |
| 724 | Ras Kankadya | -6.73 | 39.28 | CR Terrace | HTL | $7.9 \pm 1.1$ | Chronostrat. | 1 | 1 |
| 731 | Assumption Is. | -9.74 | 46.51 | Tidal inlet | NR | $4.5 \pm 2.5$ | U-Series | 2 | 3 |
| 733 | Assumption Is. | -9.74 | 46.51 | Lagoon | NR | $2.5 \pm 2.3$ | U-Series | 2 | 3 |
| 734 | Assumption Is. | -9.74 | 46.51 | CR Terrace | NR | $6.6 \pm 2.3$ | U-Series | 2 | 3 |
| 736 | Glorieuses Is. - I.d.Lys | -11.52 | 47.38 | CR Terrace | HTL | $5.5 \pm 2.3$ | U-Series | 2 | 3 |
| 937 | Praslin - PR1 | -4.34 | 55.72 | SW CR | MSL | $2.9 \pm 0.4$ | U-Series | 3 | 3 |
| 938 | Praslin - PR4 | -4.34 | 55.72 | SW CR | MSL | $8.4 \pm 0.4$ | U-Series | 3 | 1 |
| 939 | Praslin - PR7 | -4.34 | 55.73 | SW CR | MSL | $5.4 \pm 0.4$ | U-Series | 3 | 3 |
| 940 | Curieuse | -4.28 | 55.73 | SW CR | MSL | $7.4 \pm 0.4$ | U-Series | 3 | 3 |
| 941 | La Digue | -4.37 | 55.83 | CR Terrace | MSL | $2.4 \pm 0.4$ | U-Series | 3 | 3 |
| 949 | Lembetabe | -24.79 | 43.95 | CR Terrace | NR | $3.3 \pm 1.7$ | U-Series | 3 | 3 |
| 3638 | Choisy | -20.49 | 57.37 | CR Terrace | NR | $11.7 \pm 1.7$ | U-Series | 1 | 2 |