# Peer review of "Last Interglacial sea-level proxies in East Africa and the Western Indian Ocean"

_Earth System Science Data, 2020_

## Referee Comment (RC1) · Anonymous Referee #1 · 6 Jan 2021

Boyden et al., "Last Interglacial sea-level proxies in East Africa and the Western Indian Ocean"

Summary: Boyden et al provide a nice compiled dataset of Last Interglacial sea level indicators for the eastern African margin. The authors provide a nice, albeit cursory overview, of the field sites and sea-level indicators (I did appreciate this brief site/stratigraphic descriptions and figures). Currently manuscript is lacking in precision (i.e., complete and detailed methodological information) and the database is incomplete in many sections (i.e., missing values). These are simple to address, and are requisite for the dataset to be useful, significant and have longevity. This should be a useful dataset for researchers but does need a clearer explanation of age determination and a very short statement regarding the rationale for a "landform" approach (it's

not that I disagree, rather this should be more explicit for no-specialist end-users). Subject to the comments below, this manuscript and accompanying data would fall within the scope of the journal and make a good contribution to the sea-level and palaeoclimate communities.

MAIN COMMENTS (Manuscript): Insufficient, detailed description of methodology, in particular how elevation measurement and age errors were dealt with. As this is a stand-alone paper to accompany the data, you should include a summary of your methods (and it can be very brief) to reassure readers what data quality control has been done and to confirm that users are able to compare like-with-like. I found this at the very end of the document, and Section 5.4 should be moved to earlier in the manuscript (i.e., before the discussion of the sites).

Currently it is unclear whether the ages in the data are (1) comparable, or (2) are reliable. For example, comparability of the U-series dates: a) are all the ages recalculated assuming a closed system and using the same decay constants? b) do they include the decay constant error? Given that these fields are blank in the database ("RSL from single coral" sheet) – I take it not? Why not? c) Are they benchmarked (e.g., to 1950), or are they reported w.r.t. the year of measurement?

Rectifying this would only require a couple of paragraphs (max.) to the manuscript, as well as completing/tidying up the database.

With regards to the second (reliability), screening criteria are not discussed, despite their widespread use within the community. Establishing a reliable age is crucial for our understanding of sea levels during the Last Interglacial, and yet this is not dealt with in sufficient detail in sections 3 or 5 of the manuscript. Given that this is a paper/dataset concerns the Last Interglacial, but many of the ages quoted in the text (and in the dataset) are outside of the canonical age for the Last Interglacial (and MIS 5e), a very short discussion of age reliability is needed, particularly to help non-specialists appreciate some of the subtleties of the stratigraphy and age data (screening only

mentioned in passing around line 349).

Further, some clarification is needed on age determinations (e.g., U-series dates) and RSL indicators, for example the discussion of the Seychelles data. You need to specify how these replicate ages have been averaged (and screening criteria to give "accepted ages", line 283) to give the age of the unit.

Tectonic setting – it would be very useful to stress that most(?) (hard to tell from the database, data largely missing and this is known rather than unavailable) of the sites are tectonically stable, or to highlight those that are considered largely stable within your short site summaries in the manuscript.

The inclusion of the Gulf of Aqaba (Red Sea, Bar et al 2018, Yehudai et al 2017) within the geographic region is curious - what is the rationale for this given the different (tectonic and oceanographic) setting of the Red Sea? The region is also not discussed in the manuscript. It's fine, but the you have not included several key studies from the region on the Last Interglacial terraces (dates and elevation). Why are these not also included? An oversight perhaps, especially since the manuscript lays out the historical context for many of the study sub-regions. I appreciate that the dating of these reefs is difficult (they are often diagenetically altered) but it is curious that some of these studies are included (i.e., in the north, Bar, Yedudai, all highly recrystalised) and others not. Can you explain? Please include, for example, the Eritrean (Walter et al., 2000; Bruggemann et al., 2004), Egyptian (Plaziat et al., 2008, 1998, 1995), and Yemi (Al-Mikhlafi et al., 2018) Red Sea Last Interglacial terraces (see also the references within Lambeck et al., 2011), as well reference the marginal basin method (Red Sea) record (Siddall et al.,2003, 2004; Rohling et al., 2008, 2019; Grant et al., 2014). The latter doesn't need discussing, since it won't be included in your database, but it should be referenced if this region is included in the current compilation. Given the difficulty in unraveling the (potential) tectonic and age difficulties of the preserved fossil terraces, I would simply remove the Bar and Yehudai studies from your compilation.

Walter, R. C. et al. Early human occupation of the Red Sea coast of Eritrea during the last interglacial. Nature 405, 65–69 (2000).

Bruggemann, H. J. et al. Stratigraphy, palaeoenvironments and model for the deposition of the Abdur Reef Limestone: context for an important archaeological site from the last interglacial on the Red Sea coast of Eritrea. Palaeogeogr. Palaeoclimatol. Palaeoecol. 203, 179–206 (2004).

Plaziat, J.-C., Reyss, J.-L., Choukri, A. & Cazala, C. Diagenetic rejuvenation of raised coral reefs and precision of dating. The contribution of the Red Sea reefs to the question of reliability of the Uranium-series datings of middle to late Pleistocene key reef-terraces of the world. Carnets GeÌ₳ologie / Notebooks Geol. 4, 2008/04 (2008).

Plaziat, J.-C. et al. Mise en evidence, sur la cote recifale d'Egypte, d'une regression interrompant brievement le plus haut niveau du dernier interglaciaire (5e); un nouvel indice de variations glacio-eustatiques a haute frequence au Pleistocene? Bull. la SocieÌ₳teÌ₳ GeÌ₳ologique Fr. 169, 115–125 (1998).

Plaziat, J.-C. et al. Quateranry changes in the Egyptian shoreline of the northwestern Red Sea and the Gulf of Suez. Quat. Int. 29/30, 11–22 (1995).

Al-Mikhlafi, A.S., Edwards, R.L., Cheng, H. Sea-level history and tectonic uplift during the last-interglacial period (LIG) inferred from the Bab al-Mandab coral reef terraces, Yemen. J. Afr. Earth Sci. 138, 133-148 (2018).

MINOR COMMENTS (Manuscript): Line 91: strange phrasing, unclear what you mean by "...external irrepoducibility that can be puzzling high...". Please clarify and consider rephrasing.

Line 212: can you explain the discrepancy between the elevation reported in the original publication (i.e., + 10 m) and that given in your database? This just needs a few words of clarification as to why the community should use your revised elevation for this indicator.

Line 141: terraced near Merka – is this thought to be of Last Interglacial age? What's its elevation, and reference for the stud- Carbone and Accordi (2000)? Please clarify.

Line 184: fix the "(missing citation)" in the text.

Lines 192-3: not sure I follow the logic of this sentence about erosion surface age and erosion rates now and during the Last Interglacial – could you clarify, please.

Lines 220 to 222: remove "slight" from "slight issues", and insert "of the age" to "underestimation of the age of the aeolianite sedimentation". Is the inference here that the notch is therefore older than MIS 5e? Please clarify.

Lines 222: Is there any other useful information in the Hobday (1977) work – are they thought to be Last Interglaical? What elevation?

Line 320: add age given in Veeh (especially as there is only one)

Line 364: note, a fall in sea level was also suggested by Israelson and Wolfarth, (1999).

Figure 2 caption: granitic does not need to be capitalised.

MAIN COMMENTS (Data): Missing values: A considerable number of the fields are blank, including the basic site descriptions ("Nation", "Region") – is this because this data doesn't exist (e.g., % calcite determinations for the U-series ages), not applicable (e.g., uplift rates for stable locations), or incomplete data entry (e.g., blank "indicator descriptions" in the "RSL proxies" sheet, "Screening", "Location" ,"Site" in the "U-series (corals)" sheet). For users, it's vital to know which of these (not exist, not applicable, incomplete) these blanks are, especially as it could have an impact on how data is 'seen' for subsequent data analysis (e.g., training and validation in machine learning in R, Python etc.). As the author of this compilation, end users will rely on you to be clear as to whether these blanks are meaningful (rather than just incomplete data entry) and to stipulate what that meaning is. Please consider this carefully (sentinel i.e., -9999 or masking i.e., none, null – missing data or NA – not available, and NaN – not a number, recognised by most systems - might help but would need to be documented somewhere – project schema perhaps?), AND address those that arise from incomplete data entry (location, tectonic setting etc.).

(Re)calculated ages?: (see also previous comments) within the database, it is apparent (only after some digging) that some of the ages have been recalculated and others not (no information given in the manuscript); there is a mix of originally reported ages (some of which are detrital Th, or open system corrected) and recalculated (closed system?) ages. This inconsistency is confusing to the user, especially as this is not dealt with in sufficient detail in the accompanying manuscript. At the moment, non-specialists would find it difficult to decipher which age to use (and how reliable that age is) from the various sheets in the spreadsheet (even in the "Summary of RSL datapoints" it's unclear). Similarly for the age reliability (see comments above), there is a very opaque mention of a "flexible protocol" in the "Screening" column of the "RSL from a single coral" sheet of the database but no details as to what this refers to. Please clarify. You need to be very careful on this point to ensure the utility of your compiled dataset, and reduce the potential for confusion (particularly for non-specialists). One way in which you could deal with these concerns is to include within your "read me" sheet, or as a separate sheet or appendix to the manuscript, a table which describes in detail all the fields within the database..? That way, this becomes a stand-alone piece of work that has enough detail, without burdening the non-specialist with unnecessary detail.

You might consider some ranking system for the reliability of the indicator (cf. Shennan), and this is what you seem to have in the "AK" and "AL" columns of the "Summary" sheet, but why is the data entry incomplete? Where is the information on these criteria (no mention in the datafile, nor the manuscript)? End users currently have no idea what the numbers (the scale is hidden in a footnote of table 4) in these fields relate to. This needs addressing. Is there some over-arching schema from the WALIS project that can be referenced here and in the manuscript (ditto age recalculation)? If not, it might be worth considering producing one given that it would provide a permanent object

(doi?) to which you could refer in subsequent publications.

Consider adding a "tectonic setting" field to the summary (see comments in section above). This is vital information, and it was excruciating to have to flick between the various sheets to find the info, and even then it was largely missing (i.e., incomplete data entry) in the "RSL proxies" sheet. Please complete the data fields and consider adding this field to your summary.

The "chronostratigraphy" sheet is a nice addition.

I am not qualified to comment on the luminescence data.

Some language may be unclear for non-native English speakers, for example, "sketchy" (I grasp what you are driving at, but there is also an implicit value judgement) in elevation comments. Consider revising to e.g., "uncertain" or "unclear".

Journal requirements: Manuscript: Data/methods new: no but appropriate (although needs more details in the paper and accompanying metadata) Potential for reuse: yes - high Methods described in sufficient detail: no but only requires only relatively small additions to the manuscript Refs appropriate/missing: yes; suggested additional references for the Red Sea region (if retained) Structure: Mostly clear and well written (only a couple of instances that may need some reworking, see comments above). Would recommend moving section 5to earlier in the manuscript and adding more methodological details.

Data quality: Accessible (i.e., author provided identifier): yes Complete: no Error estimates and sources of error discussed: no (see comments above re description of methods) Are methods/data processing state of the art: yes Data set useful and significant: yes

Article and dataset: Inconsistencies, problems, errors: The treatment of U-series dates is currently opaque to end-users (reliability, averaging etc.). This should be dealt with in the manuscript (discussion can be very brief and methodological) and noted within

the "READ ME" or accompanying metadata. Ease of reuse (i.e. format/info conducive to statistical testing etc.): multiple sheets in the spreadsheet is not ideal for reuse, and it takes some time to orient yourself as to what data is where (and what each of the fields are – perhaps add simple table/appendix with a detailed description of each of the fields). This might be an artefact of the fields called in the database when the data is extracts. It's not fatally flawed but it's not intuitive or easy to use. High quality dataset: yes (mostly)

Presentation quality: Data usable in current format and size: yes Metadata appropriate: no, requires further (brief) explanation

Rating (1 to 4, excellent to poor): Manuscript 2 (good but needs some minor revisions); Data 2 (good but needs some minor revisions) Uniqueness: part of a fantastic larger project to collate data for the Last Interglacial building on older compilations and including new regions and data. Usefulness: extremely useful Completeness: mostly complete but some of the fields within the database are incomplete and should be addressed. This is a relatively small dataset – could you combine with the Indian Ocean data, or that for southern Africa? (note, this comment is here as the journal criteria asks about 'salami-slicing' of data).

---

## Referee Comment (RC2) · Anonymous Referee #2 · 8 Jan 2021

Boyden et al. present a compilation of published last interglacial sea-level markers as a component of the larger WALIS project. I have a high degree of admiration for this project and it should prove an extremely useful resource for sea-level scientists and those in other fields in future. Boyden et al. do a good job of mining the literature for references to sea-level markers in a region in which sea-level markers are under-documented. My main concern is that the methods and data are insufficiently and inconsistently documented in the manuscript, leading the reader having to work out how values were calculated and interpret justification. This criticism notwithstanding, the database is very thorough and I am very supportive of this kind of work. It falls clearly within the remit of the journal, making for a very useful contribution for researchers across the Earth Sciences.

[Figure]

Major Comments

One of the authors' stated goals is to standardise reporting of sea-level markers so that they are comparable. In practice, this approach means categorising sea-level markers, quantifying uncertainties in measurements and indicative range, and establishing the elevation of modern equivalents. This undertaking is challenging as the authors note since many authors, prior to the advent of GPS, do not adequately report their height measurements. This goal is a good one, but it is unclear how successful the authors have been because their documentation of this procedure is inadequate and inconsistent. The companion manuscript for this excellent database needs to very clearly and methodically spell out what the authors did to generate the database. For example the authors state that "in the literature we surveyed, it was often unclear how most datums were established", but in the description of each site, there is rarely an explanation of how the authors established their own datum or relative water level (RWL). Although this information is provided in tables and in the database, it is often very difficult and time consuming to cross reference everything. For example, I often really struggled to ascertain how site-specific RWL and indicative range (IR) values are estimated based upon the description in Table 1. The authors should strongly consider including systematic descriptions and methodological information for every measurement in the text.

A second issue is that it is difficult to determine at times to whose PRSL estimates the author are referring, or indeed whether they are referring to PRSL estimates or simply a height above an (often unspecified) datum. This way of writing is very confusing but very easy to fix! I would strongly recommend that the authors return to the text and ensure that every description includes: 1. Reference to the type of sea-level marker, its accompanying RWL, IR and a clear justification based upon the measurements and observations made in the primary literature. 2. The height reported in the primary work and above which datum (if defined, and stated if it is not). 3. The authors' own, updated PRSL estimate based upon the measurements that have been clearly spelled out.

A more detailed, general methodological description and explanation of general diffi-

culties/uncertainties should be included. This change will mean moving Section 5.4 into Sections 2 & 3 and expanding. For example, there is no discussion of specific problems with U-series dating. This problem is extremely important! There should be a short description of how authors screen their samples (calcite %, original U ratio etc.). There should also be a description of the problems of open-system behaviour. In general, this point is poorly addressed in the manuscript. There are studies cited which use open-system age-determination schemes which are not referred to (e.g. Stephenson et al., 2019). These issues should be highlighted in the detailed site descriptions as well. Again, much of this information is buried in the spreadsheet but it should be clearly spelled out in the text as it is vital for non-specialists.

Below are general comments that I wrote as I read the manuscript and explored the database, which are followed by detailed line-by-line comments in the text.

General Comments

Figure 2 – It would be useful to have this map labelled with places described in Section 4. E.g. I can't find Sanaag on the map! I think it is labelled as the Gulf of Aden.

Section 3 – There is no mention in this section of the effects of alteration of samples by diagenetic process etc. This problem is a significant one and can lead to much larger, and ill-defined uncertainties than those quoted. I think this section also needs some description of open-system modelling where there is evidence of open-system behaviour (e.g. due to an original U ratio that differs from that expected for sea water).

Section 2 and Section 4 have pretty much the same heading but one is introductory and the other includes the detailed site descriptions. Is there a way to rationalise this structure? Section 2 maybe should be called "Paleo Relative Sea Level Determination"?

What is the logic behind which study sites get a Figure? I think these Figures are great to include, but it seems a little bit random which ones are included and which

are not. For example, Why are figures not included for Stephenson et al (2019) and Dutton et al (2015) if these are the two high-quality sites, as presented on Figure 2b&c? Similarly why are photographs included for some locations and not others? Obviously photographs may not be available for some sites, but it seems sensible to include photos from Stephenson et al (2019) and Dutton et al (2015) since they are the high-quality locations.

Section 4 - For this paper to be an excellent companion to the database much greater description is needed. At the moment the reader has to dive into each paper to find the details of the field work. A few sentences of concise and consistent description for each study would help enormously. In general the data are often only partly reported. The reporting system needs to be more systematic in the text so that the reader can extract all of the information that they need without looking up the sample numbers in the spreadsheet all the time, which I found quite frustrating. If a user is looking for why a particular datapoint might be an outlier, it is going to be a torturous process at the moment when all the information could be in the text. Sometimes ages are reported but not elevation. Sometimes elevation is reported as recorded by the original authors and sometimes it is the authors' updated PRSL estimate that is reported. This chopping and changing makes it quite difficult to follow what is being referred to and I would strongly suggest that the authors try and make their reporting approach more consistent. This change shouldn't be hard but would help enormously! Additionally, it needs to be clear where the authors are using indicative meaning based upon the published work's modern analog data, and where they are using IMCalc. If they are using IMCalc, what are the inputs?

Section 5.1 – I wonder if there is an opportunity for the authors to conclude anything from their impressive database on these points? As the first compilation of these data it seems a shame for the authors to leave it to others to find paleo sea level signals? It is not essential in a data publication such as this one, but it seems like a little bit of a missed opportunity.

Section 5.3 – This section is extremely cursory! Woodroff et al (2015), Braithwaite et al (2000) etc. report Holocene data from the Seychelles; Stephenson et al (2019) and Battistini (various) report a few Holocene dates from Madagascar; Camoin et al (1997) report a whole suite of U-series dates from Reunion, Mauritius and Mayotte. The authors should either remove this section or add in significantly more data. The equatorial location of this region means that Holocene terraces at 1–2 m elevation are very common indeed.

Section 5.4 – this section should be removed and the discussion added to Sections 2 & 3. I think it is important the the reader has a sense of where the uncertainties come from before reading the results. This explanation also needs to be significantly expanded to describe the procedure for determining the authors' standardised PRSL, which is quite opaque at the moment – see comments above and below!

There is a data point from Mayotte in the Comores that I think is missing from the database that the authors should consider including. See Camoin et al. (1997) "Holocene sea level changes and reef development in the southwestern Indian Ocean". Coral Reefs, 16, 247-259. and references therein. There is no U-series date but I think these islands should be mentioned for competeness.

Can the data in Table 4 be in numerical order? It is incredibly difficult and frustrating to find WALIS ID# in this table. I ended up sorting the spreadsheet numerically which not all readers may have immediately to hand.

Detailed Comments

L24 – You state that Battistini's (1984) Tatsimian is "MIS 11 or 7?" yet on Figure 1 you have only MIS7. Is there a reason for this difference? Consider standardising.

L54 – The authors quote ages here but haven't done so for any of the previous locations. Is there a reason for this difference? It might be best just to introduce and cite the authors here and then quote ages in the detailed description later.
L65 – If this database is to be used by non-experts, then it would be helpful to have RWL, IR and indicative meaning defined for the reader/user.

L73 – just the latter half of the 20th Century or also in the early half? In my experience there is very little information from either.

L103-104 – More description of methods needed here. Since this manuscript is a data publication, it is useful to have all of the data processing information in the text alongside the database. How does IMCalc work? A short paragraph stating your approach and what this software does would help hugely in interpreting the updated PRSL values that presented in Section 4!

Section 4.1.1 – Label Sanaag on map (Figure 2).

L115 – What type of transect? Topographic? How were these transects collected? From satellite DEM? Or from a ground survey? More detail needed.

L115-6 – Do you mean for this study they were derived from Google Earth or in the original study?

L120 – State that this age is a U-series age – don't make me have to look up the dating method in the table every time!

L121-123 – Is this difference in height because the authors have altered the height based upon re-interpretation of the indicative meaning? It isn't currently clear from the text so the authors should state what has caused the change in height and re-reference the original publication.

L123 – maximum age – why maximum? Is this stated because ages generally get older with alteration in open-system conditions? It would be nice to have this point clarified. If it is due to this open-system issue then it would be worth talking about why these ages are maxima in Section 3.

Section 4.1.2 – Label Banaadir on map (Figure 2).

L135 – Again, do the authors mean, "we calculate the PRSL to be..."? I think the active voice should be used for the parts where the authors have altered the PRSL since it clearly demarcates what is their work and what is in earlier publications. Where PRSL values have been calculated or updated, I think it is important to explain why they have changed. It is not clear why the authors think that "4 m" is not a correct value of PRSL from the text. The reader needs to know why the value presented here of 3.8 /- 3.9 m is better and how the uncertainty was calculated. This issue is addressed in the database, but I think the point of this companion publication should be to make these important methodological points crystal clear and interpretation of the indicative range and relative water level justified. Is it useful to report PRSL to a greater degree of precision than the initial authors' height estimate? (1sf vs 2sf)?

L138-139 – How is it determined if there is no description? Is this determined by the earlier authors or by the current authors? What height measurement from the publication, RWL, IR etc. were used to calculate this value?

L153-155 – This outlining of the open-system behaviour needs to be mentioned earlier in Section 3 and its importance discussed for interpreting dates. It is good to mention it again here but the open-system problem and original U ratio needs to be introduced in Section 3 where dating is described. It is a primary problem in U-series dating.

L166 – The authors should state what the PRSL is that is concluded in the database. This would save the reader having to go and find it!

Section 4.3 – Are there no elevation estimates or dates in Tanzania? If not I think this should be stated.

L193 - "We extract PRSL..." - how do the authors extract this PRSL? What are the geomorphic features that are used to calculate this sea level? Again, I know this is partly in the database but it needs explaining and the sea-level markers describing in the text for completeness. Often I have to take the authors' word for a lot of things at the moment.

Section 4.3.3 – Location of Dar Es Salaam needs to be on the map in figure 2.

L200 – How is this value calculated? More details needed.

L208 – "See below" – where? Section cross-reference needed.

Section 4.4.1 – Add location to map.

L210 - "is comprised of"

L212 – Please add in the elevation estimates that are in the spreadsheet (5.5 +/- 1.37 m I think).

L220 – This is the same WALIS ID# as reported in the previous section (L213) for sample AR-06-003-001. Is this correct!? I can't check because there is no field for original sample number in the database – maybe this would be a useful addition? Please put Maputo on the map.

L229 – please put Antsiranana/Diego Suarez on the map.

L242 – How does the elevation determined by Stephenson et al (2019) translate to the new PRSL? What has been changed in the current publication? It looks like the authors are using a RWL of -1.44 m according to the database, where does this value come from? These details need explaining systematically for every site. Additionally, the database says that the PRSL = 10.74 +/- 1.36 m, but the manuscript says 10.3 +/- 1.6 m. Is this a mistake?

L243 – this is the value reported by the original authors, but what is the value that has been determined in the present work? I think 8.22 +/- 1.38 m according to the database. Please be consistent in reporting these data in the companion paper because it is very difficult to understand what the elevation estimates are referring to.

L244 and onwards – The ages quoted here from Stephenson et al (2019) are open-system ages which should be noted! The authors also report conventional ages. Please check reporting of all other studies for whether they are using open-system

or conventional U-series methods and highlight this in the text and in the database.

L251 – Again what explains the difference between the original authors' height estimates and the PRSL estimate? I presume the consistent 1 m difference in the central estimate is due to the difference between MLWS used by Stephenson et al (2019) and some other datum, but in the database the RWL is stated to be -1.45 m, not -1.0 m...? Again PRSL in the database is 4.13 +/- 1.4 m, but in the text is 3.8 +/- 1.56 m. Why? I am confused! What accounts for the different (and variable) uncertainties between this work and that of Stephenson et al (2019)? is it just the extra uncertainty in IR? What creates the uncertainty in IR? What is the merit in reporting an updated PRSL to greater precision (3 sf) than the primary authors (2 sf)?

L261 – is this the value given by the original author or in the database? In the database it seems to be 3.28 +/- 1.68 m? How is this value arrived at and why is it not written in the text while it is in other sections?

L278 – I am a bit confused here, because I thought Dutton et al (2015) used MLWS specifically because that gives them the best estimate of PRSL? I understand that they also state that their corals can grow at up to 2 m below MLWS, but they deliberately pick MLWS because many corals grow up to this height on the reef flat. It is fine to add in this extra -1.0 m and the IR estimate associated with this value, but it needs to be explained! Is this range chosen because Dutton et al (2015) quote it, or is it chosen because this value is a standard value used for all of these types of data when calculating the updated PRSL? E.g. Stephenson et al (2019) also use MLWS for reef-flat corals but the RWL used for those data is about -1.44 m (in the spreadsheet at least, it is -1.0 m in the text – see above) Why? I can't marry these differences up with the RWL and IR quoted in Table 1. Is it because of tides/weather or something else? These questions apply to all data – I am just picking up on it here because these are papers with which I am familiar.

L300 – what is the chronological limit?

L301 – More information is needed, it is not clear how this 8 m estimate translates to the PRSL estimates that the authors report here.

L345 – add U to 234/238 ratio. This section talks about U ratios but this wasn't addressed in Section 3. please add discussion of this important issue to Section 3.

L348 - ditto

L351 – is this a screening process established by the referenced authors, or in this contribution? It is not clear from the text. Is it based upon XRD or upon original U ratios?

L387 – how do these best judgements work? Where these judgements are applied they should be written down and thoroughly described in the text as well as in the database if this report is to be a useful and more verbose description of the methods that will accompany the database.

Comments on Database

This is an excellent resource and is extremely thorough. I have a few suggestions that may help improve the presentation.

Why are some description fields empty?

What is the recalculated U-series age? Explanation of this value is essential! I presume this is recalculated from the U/Th concentrations reported by the authors? Please state in the manuscript text what this recalculated age is, it is not mentioned currently I don't think. It is also essential to report where original publications quote conventional ages, open-system ages and where they report both open-system ages and conventional ages.

I think the README could be expanded so that users can better understand the various columns. E.e. the sheet "U-series (Corals)" extends from column A to column DJ, but the README tab has only a sentence of information.

[Figure]

Minor points

L4 – comprised of/composed of

L14 – remove comma after spreadsheet

L31 – prevent many of these early studies from being included...

L46 – emergent, well developed. . .

L60-61 – Incorrect citation style, needs paraentheses.

L74 – the studies that we compiled

L91 – puzzingly high... - consider rephrasing.

L109 – However, only. Change to "Only". References for these reports?

L147 – incorrect citation stayle, remove parentheses.

L179 – reference for recrystallisation.

L184 - "Missing citation"(!)

L194 - "datings" - change to ages or similar. Sentence generally needs rephrasing, missing subject pronoun.

L221 – indicate

L226 – Emergent

L251 – Irodo.

L257 – Emergent

L263 – Tidy up referencing so that year is not in double parentheses.

L282 – 7.4 +/- 0.2 m above MLWS.

L309 – disconformibly

L350 – Remove "to the far extent of the island group..."

L368 - "in the literature"

---

## Author Comment (AC1) · 16 Feb 2021

**Response to Comments by Referee #1 to "Last Interglacial sea-level proxies in East Africa and the Western Indian Ocean"**

Patrick Boyden[1], Jennifer Weil-Accardo[2], Pierre Deschamps[2], Davide Oppo[3], and Alessio Rovere[1]

[1]MARUM - Center for Marine Environmental Sciences, University of Bremen, Germany
[2]Aix Marseille Université, CNRS, IRD, Collège de France, CEREGE, France
[3]Sedimentary Basins Research Group, School of Geosciences, University of Louisiana at Lafayette, USA

**Correspondence:** Patrick Boyden (pboyden@marum.de)

**1  Summary**

We thank Anonymous Referee #1 for their thorough review of our manuscript. In the following, we answer the main comments for the manuscript and database, as well as their corresponding minor comments. The original reviewer comments are in italics while our response is in plain text and the adjusted manuscript text is indented.

## 2  Main Comments (Manuscript)

*Insufficient, detailed description of methodology, in particular how elevation measurement and age errors were dealt with. As this is a stand-alone paper to accompany the data, you should include a summary of your methods (and it can be very brief) to reassure readers what data quality control has been done and to confirm that users are able to compare like-with-like. I found this at the very end of the document, and Section 5.4 should be moved to earlier in the manuscript (i.e., before the discussion*
*of the sites).*

Here we have followed Referee #1's advice and have moved Section 5.4 earlier, into the methodology section of the manuscript. We have also expanded the section to include the quality evaluation table used throughout WALIS for both elevation and age. Below is the text we added, we hope that this addresses properly the issue raised by the referee.

> The aim of WALIS is to provide the most objective evaluation of PRSL data as possible. It therefore must be
> explicitly noted that each data set is evaluated by a set of quality control standards that are used throughout the
> WALIS database (Table 1, Rovere et al. (2020)). For the most part, elevation measurements were stated in plain
> language by the original authors, without describing in detail neither measurement methodologies nor measurement errors. We have therefore applied our best estimate errors in these cases based on the standard accuracy of the survey methodologies employed by the original authors. When we have done so, we mention this in our evaluation
> of the RSL Proxy Quality inside the database.

As discussed previously (Section 2.3), $^{238}$Th/U ages are reliant upon the technique and transparency of metadata. While many earlier studies briefly refer to the methodology used, they often provide little, if any, analytical meta-data. Within the database, we have accepted all $^{238}$Th/U ages as reported by the original authors and have only reported recalculated ages from Chutcharavan and Dutton (2020) which utilize $^{234}$U and $^{238}$Th decay constants from Cheng et al. (2013). Each chronological constraint has been rated using the common guidelines provided in the WALIS documentation (Table 2). Additionally, we have reported open-system ages for samples that are derived from mollusks (e.g. *T.Gigga*), which are widely accepted as providing inconsistent $^{238}$Th/U age reliability (e.g. Ayling et al. (2017)) and therefore have been assigned a Marine Isotopic Stage designation rather than an outright age. Data with quality higher than 4 (good) are from the most recent studies within this region and are those who have adopted more rigorous sample screening procedures and have access to the most recent advances in mass-spectrometry (e.g. MC-ICPMS).

**Table 1.** Quality rating guidelines used for evaluating PRSLs. Exported from WALIS (Rovere et al., 2020)

| Description | Quality Rating |
| --- | --- |
| Elevation precisely measured, referred to a clear datum and RSL indicator with a very narrow indicative range. Final RSL uncertainty is submetric. | 5 (excellent) |
| Elevation precisely measured, referred to a clear datum and RSL indicator with a narrow indicative range. Final RSL uncertainty is between one and two meters. | 4 (good) |
| Uncertainties in elevation, datum or indicative range sum up to a value between two and three meters. | 3 (average) |
| Final paleo RSL uncertainty is higher than three meters. | 2 (poor) |
| Elevation and/or indicative range must be regarded as very uncertain due to poor measurement/description / RSL indicator quality. | 1 (very poor) |
| There is not enough information to accept the record as a valid RSL indicator (e.g. marine or terrestial limiting). | 0 (rejected) |

*Currently it is unclear whether the ages in the data are (1) comparable, or (2) are reliable. For example, comparability of the U-series dates: a) are all the ages recalculated assuming a closed system and using the same decay constants? b) do they include the decay constant error? Given that these fields are blank in the database ("RSL from single coral" sheet) – I take it not? Why not? c) Are they benchmarked (e.g., to 1950), or are they reported with respect to the year of measurement? Rectifying this would only require a couple of paragraphs (max.) to the manuscript, as well as completing/tidying up the database.*

Recalculated ages were contributed to the database by Chutcharavan and Dutton (2020). In order to clarify this, we have added the following to Section 2.6 Uncertainties and Data Quality:

Within the database, we have accepted all $^{238}$Th/U ages as reported by the original authors and have only reported recalculated ages from Chutcharavan and Dutton (2020) which utilize $^{234}$U and $^{238}$Th decay constants from Cheng et al. (2013).

**Table 2.** Quality rating guidelines used for evaluating age information. Exported from the WALIS (Rovere et al., 2020)

| Description | Quality Rating |
| --- | --- |
| Very narrow age range, e.g. few ka, that allow the attribution to a specific timing within a substage of MIS 5 (e.g. 117 ± 2 ka) | 5 (excellent) |
| Narrow age range, allowing the attribution to a specific substage of MIS 5 (e.g., MIS 5e). | 4 (good) |
| The RSL data point can be attributed only to a generic interglacial (e.g. MIS 5). | 3 (average) |
| Only partial information or minimum age constraints are available. | 2 (poor) |
| Different age constraints point to different interglacials. | 1 (very poor) |
| Not enough information to attribute the RSL data point to any pleistocene interglacial. | 0 (rejected) |

*With regards to the second (reliability), screening criteria are not discussed, despite their widespread use within the community. Establishing a reliable age is crucial for our understanding of sea levels during the Last Interglacial, and yet this is not dealt with in sufficient detail in sections 3 or 5 of the manuscript. Given that this is a paper/dataset concerns the Last Interglacial,*

*but many of the ages quoted in the text (and in the dataset) are outside of the canonical age for the Last Interglacial (and MIS 5e), a very short discussion of age reliability is needed, particularly to help non-specialists appreciate some of the subtleties of the stratigraphy and age data (screening only mentioned in passing around line 349).*

We have looked to answer this shortcoming of the manuscript by adding additional clarification to the Dating Techniques subsection as well as expand on the screening fields within the database

*Further, some clarification is needed on age determinations (e.g., U-series dates) and RSL indicators, for example the discussion of the Seychelles data. You need to specify how these replicate ages have been averaged (and screening criteria to give "accepted ages", line 283) to give the age of the unit.*

We have clarified that the averaging was done by the original authors.

Each sample was sub-sampled in triplicate and ages have been variance-weighted averaged from the sub-samples(Dutton et al., 2015).

*Tectonic setting – it would be very useful to stress that most(?) (hard to tell from the database, data largely missing and this is known rather than unavailable) of the sites are tectonically stable, or to highlight those that are considered largely stable within your short site summaries in the manuscript.*

We had originally omitted Tectonic setting because there was no vertical land movement (VLM) stated in literature that was independent from sea-level. This requirement needs to be met when entering metadata into the database in order to make the Tectonic section available. We have gone back through each PRSL proxy and added the tectonic setting (without VLM) when mentioned by the original authors. We have also added an introduction to tectonics of the region in our methodology section (see below). For each site that is not considered stable, we briefly mention whether the PRSL is uplifting or subsiding.

The tectonic setting of PRSL indicators plays a significant role in their interpretation. Active faulting is found throughout the EAWIO (Figure 1a). For example, the majority of the East Africa coast is sitting atop the Somalia Plate that is slowly moving eastward as the East African Rift Zone (EARZ) slowly opens. Spreading rates in the EARZ decrease from north to south, 4.5 mm/a in Ethiopia to 1.5 mm/a along the Mozambique coastal plain (Stamps et al., 2008). While to the north, the Gulf of Aden is home to the Arabia-Danakil-Somalia triple junction.

When reported in literature, tectonic categories (stable, uplifting, or subsiding) are recorded within the database as to give the best possible picture of each sea level indicator setting. However, the magnitude of vertical land movements (VLM) are not explicitly included in the database. This is done because the VLM rates that were reported in literature tend to be derived from several different assumed eustatic sea levels from the LIG. As this is directly tied to sea level, it does not meet the strict "sea-level independent" criteria for insertion into the database.

*The inclusion of the Gulf of Aqaba (Red Sea, Bar et al 2018, Yehudai et al 2017) within the geographic region is curious - what is the rationale for this given the different (tectonic and oceanographic) setting of the Red Sea? The region is also not discussed in the manuscript. It's fine, but the you have not included several key studies from the region on the Last Interglacial terraces (dates and elevation). Why are these not also included? An oversight perhaps, especially since the manuscript lays out the historical context for many of the study sub-regions. I appreciate that the dating of these reefs is difficult (they are often diagenetically altered) but it is curious that some of these studies are included (i.e., in the north, Bar, Yedudai, all highly recrystallised) and others not. Can you explain? Please include, for example, the Eritrean (Walter et al., 2000; Bruggemann et al., 2004), Egyptian (Plaziat et al., 2008, 1998, 1995), and Yemi (Al- Mikhlafi et al., 2018) Red Sea Last Interglacial terraces (see also the references within Lambeck et al., 2011), as well reference the marginal basin method (Red Sea) record (Siddall et al.,2003, 2004; Rohling et al., 2008, 2019; Grant et al., 2014). The latter doesn't need discussing, since it won't be included in your database, but it should be referenced if this region is included in the current compilation. Given the difficulty in unraveling the (potential) tectonic and age difficulties of the preserved fossil terraces, I would simply remove the Bar and Yehudai studies from your compilation.*

The inclusion of the Red Sea was an oversight in the database exportation process as the sites were automatically exported because the lead author had inserted them into WALIS. We have removed these sites from the newest version of the database.

**3  Minor Comments (Manuscript)**

*Line 91: strange phrasing, unclear what you mean by ". . .external irreproducibility that can be puzzling high. . .". Please clarify and consider rephrasing.*

This sentence has been removed for better flow within the paragraph.

*Line 212: can you explain the discrepancy between the elevation reported in the original publication (i.e., + 10 m) and that given in your database? This just needs a few words of clarification as to why the community should use your revised elevation for this indicator.*

The point at Zengueleme is a Terrestrial limiting point and its elevation is not explicitly stated in Armitage et al. (2006). We have therefore deduced the approximate elevation of from the depth of the sample and the elevation of the dune. This uncertainty is reflected by the elevation uncertainty of ± 5 m.

*Line 141: terraced near Merka – is this thought to be of Last Interglacial age? What's its elevation, and reference for the stud-*
*Carbone and Accordi (2000)? Please clarify.*

We have adjusted the text to better clarify that the Merka terrace elevation and age are in the database as WALIS ID# 351.

Approximately 65 km down the coast from Mogadishu, near the small city of Merka,Carbone and Accordi (2000) describe a sheltered, well-developed reef with massive corals in growth position. We calculated PRSL of +6.4 ± 1.5 m and a correlated age to the terraces to the north of between 105 and 131 ka (WALIS ID# 351).

*Line 184: fix the "(missing citation)" in the text.*

We have added the correct citation reference.

*Lines 192-3: not sure I follow the logic of this sentence about erosion surface age and erosion rates now and during the Last Interglacial – could you clarify, please.*

We have modified the text to more clearly highlight erosion rates are too low for the terrace to be of Holocene age.

Arthurton et al. (1999) argues that the erosional surface of the marine terrace is of late-MIS 5 age because necessary erosion rates for the terrace to be of Holocene age far outpace the observed modern rates on Zanzibar as well as the lack of geological evidence for rapid sea-cliff retreat (i.e. talus deposits).

*Lines 220 to 222: remove "slight" from "slight issues", and insert "of the age" to "underestimation of the age of the aeolianite sedimentation". Is the inference here that the notch is therefore older than MIS 5e? Please clarify.*

We have followed the Reviewer's recommendation to remove slight as well as adjust the text to highlight the issues in regards to the OSL age and the 2-$\sigma$ value reported.

This gives a maximum age constraint to the notch, and is therefore inferred to be of MIS 5 age. However, Armitage et al. (2006) indicate issues with the reliability of the OSL age, suggesting that this is possibly an underestimation of the age of the aeolianite sedimentation. This is highlighted by the 2-$\sigma$ of ± 24 ka.

*Lines 222: Is there any other useful information in the Hobday (1977) work – are they thought to be Last Interglaical? What elevation?*

We have added additional reference to information provided by Hobday (1977) however there is not enough information to include this information as PRSL indicators within the database.

*Line 320: add age given in Veeh (especially as there is only one)*

The PRSL indicator from Veeh (1966) was stated in the following sentence. We have modified the text for better flow.

The morphological description of the reefs was accompanied by one U-Series Alpha-Spectrometry age from Veeh (1966). This index stands between +1.5 and +2 m MSL, representing a PRSL of 3.1 ± 2.3 m at 110 ± 40 ka (VE66-012-001, WALIS ID# 427; Table 4).

*Line 364: note, a fall in sea level was also suggested by Israelson and Wolfarth, (1999). Figure 2 caption: granitic does not*
*need to be capitalized.*

We have not included Israelson and Wohlfarth (1999) within this section because they do not reference fluctuations during MIS 5e rather only a single peak followed by a fall. We have, however, added some additional mentions of fluctuations even though they are poorly constrained.

Sea-level fluctuations during the LIG, subsequent rises and falls within MIS 5, have been alluded to by several
studies in the EAWIO region. For example Montaggioni and Hoang (1988), argue for two peaks, one between 139-133 ka and another at about 123 ka based on the distribution of their U-Series Alpha-Spectrometry ages across the granitic Seychelles. Brook et al. (1996) also identify apparent fluctuations in LIG sea level. Both the 8 m terrace and 16 m terrace they identified along the northern coast of Somalia (Section 3.1.1) are both most likely from the LIG. Here, there is stratigraphic evidence that regression occurred following the formation of the 16 m terrace
before the 8 m terrace incised this alluvial unit. However, the magnitude of this fluctuation is overshadowed by two caveats: this coastal region is tectonically active and the 16 m terrace age is base on one sample (BK96-009-001, WALIS ID# 702) that Brook et al. (1996) call, "extremely questionable."

It has not been until recently that surveying methodology and chronological constraints have achieved an accuracy that enables the documentation of such fluctuations. Vyverberg et al. (2018) conducted a multidisciplinary
investigation of the Seychelles record of Dutton et al. (2015). Across multiple outcrops around the main islands, reef growth is interrupted by discontinuities within the paleo record. Vyverberg et al. (2018) argue that this interruption in coral growth is the possible result of subaerial exposure during a fall in sea level or a still stand. Braithwaite (2020) revisited Braithwaite et al. (1973) and describes evidence of variations in sea level during the LIG on Aldabra. However, both studies conclude that higher resolution dating is needed in order to confirm this
hypothesis.

**4 Main Comments (Data)**

*Missing values: A considerable number of the fields are blank, including the basic site descriptions ("Nation", "Region") – is this because this data doesn't exist (e.g., % calcite determinations for the U-series ages), not applicable (e.g., uplift rates for stable locations), or incomplete data entry (e.g., blank "indicator descriptions" in the "RSL proxies" sheet, "Screening",*
*"Location" ,"Site" in the "U-series (corals)" sheet). For users, it's vital to know which of these (not exist, not applicable, incomplete) these blanks are, especially as it could have an impact on how data is 'seen' for subsequent data analysis (e.g., training and validation in machine learning in R, Python etc.). As the author of this compilation, end users will rely on you to be clear as to whether these blanks are meaningful (rather than just incomplete data entry) and to stipulate what that meaning is. Please consider this carefully (sentinel i.e., -9999 or masking i.e., none, null – missing data or NA – not available, and*
*NaN – not a number, recognized by most systems - might help but would need to be documented somewhere – project schema perhaps?), AND address those that arise from incomplete data entry (location, tectonic setting etc.).*

We agree that this as a very important point and have amended the database to reflect whether information is not available (NA) or not reported. In some instances, though, the database structure does not allow inserting text values (e.g., where numerical values are necessary.) Thus, every blank cell in the database will have to be treated as NA. A. Rovere made a note to highlight this point in the final editorial that will describe the database.

*(Re)calculated ages?: (see also previous comments) within the database, it is apparent (only after some digging) that some of the ages have been recalculated and others not (no information given in the manuscript); there is a mix of originally reported ages (some of which are detrital Th, or open system corrected) and recalculated (closed system?) ages. This inconsistency is confusing to the user, especially as this is not dealt with in sufficient detail in the accompanying manuscript. At the moment, non- specialists would find it difficult to decipher which age to use (and how reliable that age is) from the various sheets in the spreadsheet (even in the "Summary of RSL datapoints" it's unclear). Similarly for the age reliability (see comments above), there is a very opaque mention of a "flexible protocol" in the "Screening" column of the "RSL from a single coral" sheet of the database but no details as to what this refers to. Please clarify.*

We have added a brief overview of U-Series dating within the methodology of the manuscript, including the difference between open- and closed-systems, as well as throughout the text added whether stated ages are recalculated or open-/closed-system ages. Also, we have added an explanation of the "flexible protocol" used in Dutton et al. (2015).

[revised manuscript text omitted]

*You need to be very careful on this point to ensure the utility of your compiled dataset, and reduce the potential for confusion (particularly for non-specialists). One way in which you could deal with these concerns is to include within your "read me" sheet, or as a separate sheet or appendix to the manuscript, a table which describes in detail all the fields within the database..? That way, this becomes a stand-alone piece of work that has enough detail, without burdening the non-specialist with unnecessary detail.*

A "wiki-style" page for the WALIS interface is available for download, we have added the url for this page in the about section on the Zenodo (database host) page for the database for easier reference.

*You might consider some ranking system for the reliability of the indicator (cf. Shennan), and this is what you seem to have in the "AK" and "AL" columns of the "Summary" sheet, but why is the data entry incomplete? Where is the information on these criteria (no mention in the datafile, nor the manuscript)? End users currently have no idea what the numbers (the scale*

*is hidden in a footnote of table 4) in these fields relate to. This needs addressing. Is there some over-arching schema from the WALIS project that can be referenced here and in the manuscript (ditto age recalculation)? If not, it might be worth considering producing one given that it would provide a permanent object (DOI?) to which you could refer in subsequent publications.*

We have provided the evaluation/quality control guidelines within out manuscript as exported from WALIS (see above, Tables 1 and 2

*Consider adding a "tectonic setting" field to the summary (see comments in section above). This is vital information, and it was excruciating to have to flick between the various sheets to find the info, and even then it was largely missing (i.e., incomplete data entry) in the "RSL proxies" sheet. Please complete the data fields and consider adding this field to your summary.*

While we agree with the overall idea of this point, in conversation with the WALIS team we have decided to not change columns within the database export so that all database exports within the special issue are of the same format. Nevertheless, see our additional text provided on tectonics.

*Some language may be unclear for non-native English speakers, for example, "sketchy" (I grasp what you are driving at, but there is also an implicit value judgment) in elevation comments. Consider revising to e.g., "uncertain" or "unclear"*

We have looked throughout the text and addressed the colloquial English language to suit a more international audience.

**References**

[revised manuscript text omitted]

---

## Author Comment (AC2) · 16 Feb 2021

**Response to Comments by Referee #2 to "Last Interglacial sea-level proxies in East Africa and the Western Indian Ocean"**

Patrick Boyden[1], Jennifer Weil-Accardo[2], Pierre Deschamps[2], Davide Oppo[3], and Alessio Rovere[1]

[1]MARUM - Center for Marine Environmental Sciences, University of Bremen, Germany
[2]Aix Marseille Université, CNRS, IRD, Collège de France, CEREGE, France
[3]Sedimentary Basins Research Group, School of Geosciences, University of Louisiana at Lafayette, USA

**Correspondence:** Patrick Boyden (pboyden@marum.de)

**1 Summary**

We thank Anonymous Referee #2 for their thorough review of our manuscript. In the following, we answer the main comments for the manuscript and database, as well as their corresponding minor comments. The original reviewer comments are in italics while our response is in plain text and the adjusted manuscript text is indented.

**2 Major Comments**

*One of the authors' stated goals is to standardize reporting of sea-level markers so that they are comparable. In practice, this approach means categorizing sea-level markers, quantifying uncertainties in measurements and indicative range, and establishing the elevation of modern equivalents. This undertaking is challenging as the authors note since many authors, prior to the advent of GPS, do not adequately report their height measurements. This goal is a good one, but it is unclear how successful the authors have been because their documentation of this procedure is inadequate and inconsistent. The companion manuscript for this excellent database needs to very clearly and methodically spell out what the authors did to generate the database. For example the authors state that "in the literature we surveyed, it was often unclear how most datums were established", but in the description of each site, there is rarely an explanation of how the authors established their own datum or relative water level (RWL). Although this information is provided in tables and in the database, it is often very difficult and time consuming to cross reference everything. For example, I often really struggled to ascertain how site-specific RWL and indicative range (IR) values are estimated based upon the description in Table 1. The authors should strongly consider including systematic descriptions and methodological information for every measurement in the text*

In order to clarify and streamline the manuscript, we have expanded Section 2 Methods and have included a breakdown of how PRS are calculated in database.

**Surveying Techniques**
Very few studies within the EAWIO have the express intent to establish detailed surveys of Last Interglacial (LIG)

sea-level proxies. This is especially true with respect to elevation measurements. Most surveys conducted during the 20[th] century do not report a methodology used in measuring elevations. It is not until the advent of Global Navigation Satellite Systems (GNSS) and Total Stations that surveys on many of these remote shorelines could be accurately documented. The elevation measurement techniques used in the studies that we compiled in the database are shown in Table 1. When no accuracy was given for an elevation measurement in the original study, the typical accuracy of the technique was used. Any elevation measurement must be related to a specific sea-level datum (Table 3). Unfortunately, in the literature we surveyed, it was often unclear how most datums were established (e.g. how the highest tide level was calculated at different sites). Instead authors will often state that the elevation is relative to mean sea-level or the level of highest seas. This uncertainty is exacerbated by the large variance in tides within the EAWIO, specifically in the immediate vicinity of the Mozambique Channel (Farrow and Brander, 1971; Kench, 1998). In the database, we therefore try to reflect this uncertainty within the elevation measurement for each proxy.

**Paleo Relative Sea Level Estimation**

In order to extract paleo relative sea level (PRSL) from measured elevations, the IR and RWL for the measured indicator are needed (Shennan, 1982). The IR relies upon the measurement of modern upper and lower limits of the indicator in relation to an established datum. However, few studies have thoroughly documented the upper and lower limits of the site specific modern analogue to the indicator. To supplement missing IR and RWL values, Lorscheid and Rovere (2019) introduced a reliable empirical method that uses a global dataset of wave and tide model outputs in conjunction with the morpho- and hydrodynamic formation environment of the most common sea level indicators. This methodology was then packaged into a open-access stand-alone software, IMCalc, availible at: https://sourceforge.net/projects/imcalc/ (Lorscheid and Rovere, 2019). In the database we use the upper and lower limits when given by original authors, however, the majority of upper and lower limits for our PRSL points are calculated from the IMCalc software. Once the upper and lower limits are determined, WALIS automatically calculates the IR, RWL, PRSL, and PRSL uncertainty, based on the schemes from Rovere et al. (2016). All of the PRSL elevations in the following text have been calculated from originally published survey elevations using this methodology in order to standardize their comparison.

*A second issue is that it is difficult to determine at times to whose PRSL estimates the author are referring, or indeed whether they are referring to PRSL estimates or simply a height above an (often unspecified) datum. This way of writing is very confusing but very easy to fix! I would strongly recommend that the authors return to the text and ensure that every description includes: 1. Reference to the type of sea-level marker, its accompanying RWL, IR and a clear justification based upon the measurements and observations made in the primary literature. 2. The height reported in the primary work and above which datum (if defined, and stated if it is not). 3. The authors' own, updated PRSL estimate based upon the measurements that have been clearly spelled out.*

55  For each WALIS ID# we have provided PRSL elevations except for the two terrestrial limiting points. We have modified the text to more clearly outline that all PRSLs are our calculated PRSL values. Within the database we include all the metadata available for each WALIS ID#. We fear that inserting the information asked by the reviewer (which are readily accessible in the database) would have the effect of overcrowding the MS, diluting its descriptive aim.

*A more detailed, general methodological description and explanation of general difficulties/uncertainties should be included.*
60  *This change will mean moving Section 5.4 into Sections 2 & 3 and expanding. For example, there is no discussion of specific problems with U-series dating. This problem is extremely important! There should be a short description of how authors screen their samples (calcite %, original U ratio etc.). There should also be a description of the problems of open-system behavior. In general, this point is poorly addressed in the manuscript. There are studies cited which use open-system age-determination schemes which are not referred to (e.g. Stephenson et al., 2019). These issues should be highlighted in the*
65  *detailed site descriptions as well. Again, much of this information is buried in the spreadsheet but it should be clearly spelled out in the text as it is vital for non-specialists.*

  This comment echoes the concerns of Referee 1 and have moved Section 5.4 up to Section 2 as well as expanded the discussion of issues found during age determinations. We have also added, where appropriate, whether U-Series ages are open- or closed-systems. Below is the revised Dating Techniques subsection and the Uncertainties and Data Quality subsection. To
70  help be more transparent with how we have addressed quality control, we have included the rating systems for both elevations and ages.

[revised manuscript text omitted]

**3 General Comments**

Figure Comments *Figure 2 – It would be useful to have this map labelled with places described in Section 4. E.g. I can't find Sanaag on the map! I think it is labelled as the Gulf of Aden.*, *Section 4.1.2 – Label Banaadir on map (Figure 2).*, *Section 4.4.1 – Add location to map.*, and *Please put Maputo on the map.*

Unfortunately, adding such data to the map would be impossible, due to the high clusters of points present. While adding one map for each region would be feasible, we fear it would sum up to a much longer MS. Therefore, we prepared a standalone HTML file, that we provide as annex, where the user can navigate data and samples, getting more information as they read. We hope this is an acceptable compromise.

*Section 3 – There is no mention in this section of the effects of alteration of samples by diagenetic process etc. This problem is a significant one and can lead to much larger, and ill-defined uncertainties than those quoted. I think this section also needs some description of open-system modelling where there is evidence of open-system behavior (e.g. due to an original U ratio that differs from that expected for sea water).*

As stated above, we have expanded the Dating Techniques subsection to include a brief explanation of open-system behavior as well as referenced the respective open-system model individual authors used for site specific age calculations.

*Section 2 and Section 4 have pretty much the same heading but one is introductory and the other includes the detailed site descriptions. Is there a way to rationalise this structure? Section 2 maybe should be called "Paleo Relative Sea Level Determination"?*

To address this confusion we have renamed Section 2 (now subsection 2.5) "Paleo Relative Sea Level Estimation".

*What is the logic behind which study sites get a Figure? I think these Figures are great to include, but it seems a little bit random which ones are included and which are not. For example, Why are figures not included for Stephenson et al (2019) and Dutton et al (2015) if these are the two high-quality sites, as presented on Figure 2b&c? Similarly why are photographs included for some locations and not others? Obviously photographs may not be available for some sites, but it seems sensible to include photos from Stephenson et al (2019) and Dutton et al (2015) since they are the high- quality locations.*

We did not include sketches from Dutton et al. (2015) and Stephenson et al. (2019) because these two studies particularly focus on sea-level reconstruction and when incorporating their data into our database, the morphological nature of the respective outcrops is not as relied upon as with many of the earlier studies where the PRSL proxies are very much "extracted" from geomorphological descriptions of outcrops. As for the photos, we have added only photographs taken by the authors of the MS, for which we have the rights to reproduce. The two papers cited by the reviewer are widely available and contain high-quality photos, therefore we decided not to include any photographs that might require copyright clearance.

*Section 4 - For this paper to be an excellent companion to the database much greater description is needed. At the moment the reader has to dive into each paper to find the details of the field work. A few sentences of concise and consistent description for each study would help enormously. In general the data are often only partly reported. The reporting system needs to be more systematic in the text so that the reader can extract all of the information that they need without looking up the sample numbers in the spreadsheet all the time, which I found quite frustrating. If a user is looking for why a particular datapoint might be an outlier, it is going to be a torturous process at the moment when all the information could be in the text. Sometimes ages are reported but not elevation. Sometimes elevation is reported as recorded by the original authors and sometimes it is the authors' updated PRSL estimate that is reported. This chopping and changing makes it quite difficult to follow what is being referred to and I would strongly suggest that the authors try and make their reporting approach more consistent. This change shouldn't be hard but would help enormously!*

Within the manuscript we do not focus on the field work element because the majority of RSL proxies are derived from papers that do not describe surveying techniques (summarized in Subsection 2.2 Surveying Techniques). We therefore try to focus on the standardized data in order to enable the comparison of data. As for the confusion between originally stated ages and the calculated PRSLs, we have gone through and modified the text to more explicitly tie the ages, and therefore the WALIS ID#s, the calculated PRSL values.

*Additionally, it needs to be clear where the authors are using indicative meaning based upon the published work's modern analog data, and where they are using IMCalc. If they are using IMCalc, what are the inputs?*

As stated above, we have expanded the section where we introduce the calculation of PRSL values. This includes an introduction to the IMCalc app and its inputs.

*Section 5.1 – I wonder if there is an opportunity for the authors to conclude anything from their impressive database on these points? As the first compilation of these data it seems a shame for the authors to leave it to others to find paleo sea level signals? It is not essential in a data publication such as this one, but it seems like a little bit of a missed opportunity.*

While it is tempting to draw conclusions from the data standardized in this database, this is beyond the scope of the project. It is the hope that future studies can take advantage of our database (as stated in our Future Directions section) and apply GIA corrections to draw inferences on regional or eustatic sea level.

*Section 5.3 – This section is extremely cursory! Woodroff et al (2015), Braithwaite et al (2000) etc. report Holocene data from the Seychelles; Stephenson et al (2019) and Battistini (various) report a few Holocene dates from Madagascar; Camoin et al (1997) report a whole suite of U-series dates from Reunion, Mauritius and Mayotte. The authors should either remove this section or add in significantly more data. The equatorial location of this region means that Holocene terraces at 1–2 m elevation are very common indeed.*

In deed, this section was far too cursory and had focused more on controversial deposits that have been attributed to both Holocene and the LIG. We therefore removed the section from the manuscript. Other database projects, such as HOLSEA (Khan et al., 2019), are more appropriate repositories for standardized Holocene sea-level data.

*Section 5.4 – this section should be removed and the discussion added to Sections 2 & 3. I think it is important the the reader has a sense of where the uncertainties come from before reading the results. This explanation also needs to be significantly expanded to describe the procedure for determining the authors' standardised PRSL, which is quite opaque at the moment – see comments above and below!*

Again, we have moved this subsection up in the manuscript to Section 2.6 as well as expanded the explanation of our methodologies used in determining standardized PRSLs.

*There is a data point from Mayotte in the Comores that I think is missing from the database that the authors should consider including. See Camoin et al. (1997) "Holocene sea level changes and reef development in the southwestern Indian Ocean". Coral Reefs, 16, 247-259. and references therein. There is no U-series date but I think these islands should be mentioned for completeness.*

Thank you for pointing Camoin et al. (1997) out. We have reviewed the text but cannot include it in the database as no chronological constraints are available. We do however mention it in passing as evidence to the lack of other MIS 5e RSL proxies on Mayotte.

*Can the data in Table 4 be in numerical order? It is incredibly difficult and frustrating to find WALIS ID# in this table. I ended up sorting the spreadsheet numerically which not all readers may have immediately to hand.*

This is a very good point for quality of life. We have reordered the table to reflect WALIS ID#s.

**4   Detailed Comments**

We feel that many of the below comments have been answered by the responses provided above. We therefore in many instances state, "Please see responses above."

*L24 – You state that Battistini's (1984) Tatsimian is "MIS 11 or 7?" yet on Figure 1 you have only MIS7. Is there a reason for this difference? Consider standardising.*

We have standardized the Tatsimian to reflect the usage in Figure 1.

*L54 – The authors quote ages here but haven't done so for any of the previous locations. Is there a reason for this difference? It might be best just to introduce and cite the authors here and then quote ages in the detailed description later.*

We agree with this comment and have removed references to ages from the introduction.

*L65 – If this database is to be used by non-experts, then it would be helpful to have RWL, IR and indicative meaning defined for the reader/user.*

Please see responses above.

*L73 – just the latter half of the 20th Century or also in the early half? In my experience there is very little information from either.*

We agree and have modified the text to reflect that.

*L103-104 – More description of methods needed here. Since this manuscript is a data publication, it is useful to have all of the data processing information in the text alongside the database. How does IMCalc work? A short paragraph stating your approach and what this software does would help hugely in interpreting the updated PRSL values that presented in Section 4!*

Please see responses above.

*L115 – What type of transect? Topographic? How were these transects collected? From satellite DEM? Or from a ground survey? More detail needed.*

We have added additional text clarifying that these transects were gathered from a ground survey.

> Mapping of the area was conducted using aerial photographs in conjunction with a series of four transects using altimeter measurements from the field (Brook et al., 1996).

*L115-6 – Do you mean for this study they were derived from Google Earth or in the original study?*

We have modified the text to clarify that we used the Google Earth to estimate the coordinates based on the original published map.

Coordinates of samples and terraces in the database were estimated in Google Earth from the original published maps.

235 *L120 – State that this age is a U-series age – don't make me have to look up the dating method in the table every time!*

This was an oversight by us. We have gone through the manuscript and added the type of chronological constraint when missing.

*L121-123 – Is this difference in height because the authors have altered the height based upon re-interpretation of the indicative meaning? It isn't currently clear from the text so the authors should state what has caused the change in height and*
240 *re-reference the original publication.*

We have tried to clarify this confusion by expanding on the methodology, particularly we state reference survey elevations (and their respective datums) from the original publications as well as our calculated PRSL.

*L153-155 – This outlining of the open-system behaviour needs to be mentioned earlier in Section 3 and its importance discussed for interpreting dates. It is good to mention it again here but the open-system problem and original U ratio needs to*
245 *be introduced in Section 3 where dating is described. It is a primary problem in U-series dating.*

Please see responses above.

*L166 – The authors should state what the PRSL is that is concluded in the database. This would save the reader having to go and find it!*

We have modified the text to clarify this misunderstanding.

250 The samples were taken from on top of the coral reef terrace within an elevation range of 8 - 15 m above mean sea level and have an open-system age of $120 \pm 8$ ka and a PRSL of between +14.5 m and + 21.5 m (MIS-5e, WALIS ID #s 189-192; Table 6). Groups B and C are taken from the face of the limestone cliffs 0 and 6 m above mean sea level. Group B samples come from the central to northern section of coast between Kalifi and Manda Island, and have an open-system age of $118 \pm 14$ ka and a claculated PRSL of between +9.5 m and + 14.5 m (MIS-5e to 5d,
255 WALIS ID#s 193-198; Table 6). Finally, Group C has an open-system age of $100 \pm 8$ ka and a claculated PRSL of between +8.5 m and +12.5 m (MIS-5c to 5d, WALIS ID#s 199-201) and is located along the same section of coast as Group A (Schimoni to Kalifi).

*Section 4.3 – Are there no elevation estimates or dates in Tanzania? If not I think this should be stated.*

The individual subsections of Section 4.3 Tanzania introduce the different PRSL estimates in country.
260 *L193 - "We extract PRSL..." - how do the authors extract this PRSL? What are the geomorphic features that are used to calculate this sea level? Again, I know this is partly in the database but it needs explaining and the sea-level markers describing in the text for completeness. Often I have to take the authors' word for a lot of things at the moment.*

We have modified the text to better clarify the relationship of the PRSL to the geomorphic observations.

Kourampas et al. (2015) provide a descriptive transect of the Jambiani marine terrace from which we calculate a
265 PRSL of $+11 \pm 5.1$ m for MIS 5 (WALIS ID# 212).

*L200 – How is this value calculated? More details needed.*

We have modified the text to reflect that this PRSL was derived from a coral reef terrace.

> From this the coral reef terrace we calculate a PRSL of +7.9 ±1.1 m for (WALIS ID# 724).

*L208 – "See below" – where? Section cross-reference needed.*

We have added Section cross-reference here

*L212 – Please add in the elevation estimates that are in the spreadsheet (5.5 +/- 1.37 m I think)*

This is a terrestrial limiting point and therefore there is no PRSL estimate. We have added this explanation to the manuscript to prevent this confusion.

> There is no PRSL for this formation as this is only a terrestrial limiting point.

*L220 – This is the same WALIS ID# as reported in the previous section (L213) for sample AR-06-003-001. Is this correct!? I can't check because there is no field for original sample number in the database – maybe this would be a useful addition?*

Thank you for catching this. The WALIS ID# has been updated to reflect the correct ID, 184

*L242 – How does the elevation determined by Stephenson et al (2019) translate to the new PRSL? What has been changed in the current publication? It looks like the authors are using a RWL of -1.44 m according to the database, where does this value come from? These details need explaining systematically for every site. Additionally, the database says that the PRSL = 10.74 +/- 1.36 m, but the manuscript says 10.3 +/- 1.6 m. Is this a mistake?*

This RWL value comes from the upper and lower limits derived from IMCalc, we have used this system to recalculate the PRSL. As for the difference between the database and manuscript, thank you for catching this. This is a mistake and we have corrected the value in the manuscript.

*L243 – this is the value reported by the original authors, but what is the value that has been determined in the present work? I think 8.22 ± 1.38 m according to the database. Please be consistent in reporting these data in the companion paper because it is very difficult to understand what the elevation estimates are referring to.*

We have added a following sentence with the appropriate PRSL value.

> Moving south along the eastern shoreline, near Baie des Dunes from Battistini et al. (1976), a coral reef terrace elevation from Cap Miné is recorded at +6.8 ± 1.2 m above MLWS with an age between 125.5 ± 1.8 and 136.9 ± 2.3 ka (ST18-003-001 and ST18-004-001, WALIS ID# 159, Table 6). We calculate a PRSL of 8.2 ± 1.4 m from Cap Miné.

*L244 and onwards – The ages quoted here from Stephenson et al (2019) are open- system ages which should be noted! The authors also report conventional ages. Please check reporting of all other studies for whether they are using open-system or conventional U-series methods and highlight this in the text and in the database.*

As referenced above, we have gone through the manuscript and noted when open system ages are used and when they are not.

*L251 – Again what explains the difference between the original authors' height estimates and the PRSL estimate? I presume the consistent 1 m difference in the central estimate is due to the difference between MLWS used by Stephenson et al (2019) and some other datum, but in the database the RWL is stated to be -1.45 m, not -1.0 m. . .? Again PRSL in the database is 4.13 +/- 1.4 m, but in the text is 3.8 +/- 1.56 m. Why? I am confused! What accounts for the different (and variable) uncertainties between this work and that of Stephenson et al (2019)? is it just the extra uncertainty in IR? What creates the uncertainty in IR? What is the merit in reporting an updated PRSL to greater precision (3 sf) than the primary authors (2 sf)?*

Thank you for catching this. We have updated the manuscript with the correct PRSL from the database. The inclusion of 3 significant figures is a mistake and has been corrected. As for the use of a difference in sf between the PRSL and the primary authors, this is a byproduct of the database itself and how it handles inputs for elevation as floats. We have therefore not changed these values to reflect a standardized comparison within the whole database.

*L261 – is this the value given by the original author or in the database? In the database it seems to be 3.28 +/- 1.68 m? How is this value arrived at and why is it not written in the text while it is in other sections?*

This was the original elevation provided by Battisitini, we have added the newly calculated PRSL from the database to clarify this.

The emerged reef, with large in-situ corals in growth position, was described at 1 - 2 m above MSL with an age of $85 \pm 10$ ka (WALIS ID# 949, BA76-001-001). We have calculated a PRSL of $+3.3 \pm 1.7$ m.

*L278 – I am a bit confused here, because I thought Dutton et al (2015) used MLWS specifically because that gives them the best estimate of PRSL? I understand that they also state that their corals can grow at up to 2 m below MLWS, but they deliberately pick MLWS because many corals grow up to this height on the reef flat. It is fine to add in this extra -1.0 m and the IR estimate associated with this value, but it needs to be explained! Is this range chosen because Dutton et al (2015) quote it, or is it chosen because this value is a standard value used for all of these types of data when calculating the updated PRSL? E.g. Stephenson et al (2019) also use MLWS for reef-flat corals but the RWL used for those data is about -1.44 m (in the spreadsheet at least, it is -1.0 m in the text – see above) Why? I can't marry these differences up with the RWL and IR quoted in Table 1. Is it because of tides/weather or something else? These questions apply to all data – I am just picking up on it here because these are papers with which I am familiar.*

The difference between the RWL in the Dutton et al. (2015) dataset and Stephenson et al. (2019) are different because the tidal range of the two localities are different and therefore the indicative range of coral reef terraces are different (upper limit of the IR is mean lower low water). This difference is furthered because Dutton et al. (2015) provide modern analogue data where as the PRSL data from Stephenson et al. (2019) is derived using IMCalc. As responded above, we have added a subseciton for how PRSL elevations are derived in the Methods section.

*L300 – what is the chronological limit?*

This was a misleading sentence and has been reworded to make it more clear that this is the chronological limit from the proceeding sentence.

*L301 – More information is needed, it is not clear how this 8 m estimate translates to the PRSL estimates that the authors report here.*

Please see responses above.

*L345 – add U to 234/238 ratio. This section talks about U ratios but this wasn't ad- dressed in Section 3. please add discussion of this important issue to Section 3.*

Please see responses above.

*L351 – is this a screening process established by the referenced authors, or in this contribution? It is not clear from the text. Is it based upon XRD or upon original U ratios?*

We have added clarification that this was the original publication's screening process.

*L387 – how do these best judgements work? Where these judgements are applied they should be written down and thoroughly described in the text as well as in the database if this report is to be a useful and more verbose description of the methods that will accompany the database.*

We have moved this subsection up to the methods section and have looked to clarify the elevation.

**5   Comments on Database**

*Why are some description fields empty? What is the recalculated U-series age? Explanation of this value is essential! I pre- sume this is recalculated from the U/Th concentrations reported by the authors? Please state in the manuscript text what this recalculated age is, it is not mentioned currently I don't think. It is also essential to report where original publications quote conventional ages, open-system ages and where they report both open-system ages and conventional ages. I think the README could be expanded so that users can better understand the various columns. E.e. the sheet "U-series (Corals)" extends from column A to column DJ, but the README tab has only a sentence of information.*

This comment was also brought up by Referee #1. We have not recalculated any ages, the only recalculated ages we record in the database are from Chutcharavan and Dutton (2020). We have also added within the manuscript where open-system ages are used and where closed-system ages are used.

**6   Minor Points**

We have followed the Referee's recommendations and modified the manuscript accordingly.

**References**

[revised manuscript text omitted]

---

## Author Response (AR2)

**Response to Matteo Vacchi's (Topical Editor) Minor Comments on: *Last Interglacial sea-level proxies in East Africa and the Western Indian Ocean**

Patrick Boyden[1], Jennifer Weil-Accardo[2], Pierre Deschamps[2], Davide Oppo[3], and Alessio Rovere[1]

[1]MARUM - Center for Marine Environmental Sciences, University of Bremen, Germany
[2]Aix Marseille Université, CNRS, IRD, Collège de France, CEREGE, France
[3]Sedimentary Basins Research Group, School of Geosciences, University of Louisiana at Lafayette, USA

**Correspondence:** Patrick Boyden (pboyden@marum.de)

**Summary.** We thank Matteo Vacchi's for their thorough review of our manuscript. here, we look to address each comment. The three minor comments on significant digits and LaTeX formatting errors have been addressed.

**1      Comments and Replies**

*Please double check that every point included in the database is part of the geographical area of East Africa and Western Indian Ocean.*

> We have reviewed all data within the database and can confirm that the data included is exclusively from the EAWIO region.

*The left panel figure 2 (the map) is still not very clear. Please clearly indicate the different sub-regions you described in text. Furthermore please be sure that all the point/labels are clearly visible (even if placed close each other).*

> We have expanded the map in figure 2 and explicitly identify the different regions (sub-sections) discussed in the paper, with additional location information where space allows.

*Table 2. The table 1 should be modified. Please remove last column with the references (they are already in the second column.*
*-Line 1 (coral reef terrace). Dept > Depth*
*-Line 4 (shallow water coral reef). Please uniform the tidal nomenclature, Mean Low Water Springs (see my comment for table 3).*
*Line 5 (Tidal inlet). Please specify that -0.5 and -3.5 are expressed in meters (m).*

> Thank you for the comments on this table. We have made the appropriate changes as well as some changes within the greater WALIS database environment.

*Table 3. This tidal nomenclature differs significantly from the one described in Table 1. Please uniform this along the paper by following the Woodroffe and Barlow's Chapter 11 (Reference water level and tidal datum) of the handbook of sea level research 2015.*

> Please see our reply to Table 2.